# Downregulated NDR1 protein kinase inhibits innate immune response by initiating an miR146a-STAT1 feedback loop

Zhiyong Liu[1], Qiang Qin[1], Cheng Wu[1], Hui Li[2], Jia'nan Shou[1], Yuting Yang[1], Meidi Gu[1], Chunmei Ma[1], Wenlong Lin[1], Yan Zou[3], Yuanyuan Zhang[4], Feng Ma[5], Jihong Sun[6] & Xiaojian Wang[1]

Interferon (IFN)-stimulated genes (ISGs) play crucial roles in the antiviral immune response; however, IFNs also induce negative regulators that attenuate the antiviral response. Here, we show that both viral and bacterial invasion downregulate Nuclear Dbf2-related kinase 1 (NDR1) expression via the type I IFN signaling pathway. NDR1 promotes the virus-induced production of type I IFN, proinflammatory cytokines and ISGs in a kinase-independent manner. NDR1 deficiency also renders mice more susceptible to viral and bacterial infections. Mechanistically, NDR1 enhances STAT1 translation by directly binding to the intergenic region of miR146a, thereby inhibiting miR146a expression and liberating STAT1 from miR146a-mediated translational inhibition. Furthermore, STAT1 binds to the miR146a promoter, thus decreasing its expression. Together, our results suggest that NDR1 promotion of STAT1 translation is an important event for IFN-dependent antiviral immune response, and suggest that NDR1 has an important role in controlling viral infections.

[1] Institute of Immunology, School of Medicine, Zhejiang University, Hangzhou 310058, PR China. [2] Department of Chemotherapy, Zhejiang Cancer Hospital, Hangzhou 310022, PR China. [3] Medical Science Laboratory, The Fourth Affiliated Hospital of Guangxi Medical University, Liuzhou 545005, PR China. [4] The Children's Hospital, School of Medicine, Zhejiang University, Hangzhou 310058, PR China. [5] Suzhou Institute of Systems Medicine, Chinese Academy of Medical Sciences & Peking Union Medical College, Suzhou 215123, PR China. [6] Department of Radiology, Sir Run Run Shaw Hospital School of Medicine, Zhejiang University, Hangzhou 310058, PR China. These authors contributed equally: Zhiyong Liu, Qiang Qin. Correspondence and requests for materials should be addressed to J.S. (email: sunjihong@zju.edu.cn) or to X.W. (email: wangxiaojian@cad.zju.edu.cn)

nnate immunity is the first line of host defense in resistance against pathogen infection and is activated by pattern-recognition receptors (PRRs) that recognize pathogen-associated molecular patterns[1]. This in turn induces type I interferon (IFN) expression, and the upregulation of IFN-stimulated genes (ISGs), which have a central role in intracellular antiviral defenses[2]. In addition to antiviral ISGs induction, IFNs can also induce several negative regulators that target PRRs, IRFs, or JAK/STAT, and subsequently attenuate antiviral responses, thus functioning as "brakes" in innate immunity[3]. Thus, understanding the mechanisms that limit or downregulate type I IFN production downstream signaling of pathogen recognition is critical for developing more effective antiviral therapies.

MiR146a has been shown to function as a negative feedback regulator in Toll-like receptors (TLR) signaling by targeting IRAK1 and TRAF6[4]. Further studies have identified other targets of miR146a, such as STAT1[5–7] and IRAK2[8], which are involved in the innate antiviral immune response. In addition, miR146a expression is induced by vesicular stomatitis virus (VSV) infection[8]. Given the important roles of miR146a in innate antiviral immune responses, the delicate mechanism by which miR146a is regulated after viral infection requires further investigation.

NDR1, also known as STK38, is a ubiquitously expressed and highly conserved serine/threonine kinase in the NDR/LATS kinase family. NDR1 is a critical regulator of various cellular processes, including centrosome duplication, apoptosis, and tumorigenesis[9–12]. Recently, Wen et al.[13] have reported that NDR1 inhibits TLR9-activated inflammatory responses by promoting MEKK2 ubiquitination in macrophages.

In the present study, we find that NDR1 mRNA is significantly decreased in peripheral blood samples from patients infected with respiratory syncytial virus (RSV) compared with those from healthy controls. We also find that viral infection downregulates NDR1 expression and that NDR1 knockdown or knockout impairs ISGs induction. $NDR1^{-/-}$ mice are more susceptible to viral and Listeria monocytogenes (LM) infection compared with wild-type (WT) mice. Mechanistically, NDR1 binds to the intergenic region of miR146a and compromises NF-κB-mediated miR146a transcription, thus enhancing STAT1 translation. Furthermore, STAT1 binds to the miR146a promoter, thereby further inhibiting NF-κB-mediated miR146a transcription. Together, our data demonstrate that viral infection downregulates NDR1 expression, thus counteracting the mutual inhibition between miR146a and STAT1 and resulting in decreased STAT1 expression and impaired ISG expression, thereby allowing viral escape from innate immunity.

## Results

### Viral infection-induced NDR1 reduction requires STAT1.
We first investigated NDR1 expression in peripheral blood samples from patients infected with RSV as well as healthy controls. NDR1 mRNA levels were significantly decreased in RSV-infected patient samples compared with those from healthy controls (Fig. 1a). Downregulated NDR1 mRNA and protein levels were also observed in mouse peritoneal macrophages (PMs) infected with RNA viruses, including RSV and VSV, and DNA viruses, including herpes simplex virus 1 (HSV) and murine herpes virus 68 (MHV68) (Fig. 1b, c). Moreover, NDR1 expression was decreased in human primary peripheral blood mononuclear cells (PBMCs) and various macrophages infected with VSV, including human acute monocytic leukemia cell line Thp1, murine bone marrow-derived macrophages (BMDMs) and RAW264.7 (Supplementary Fig. 1a, b). In addition, transfection of poly(I:C) and poly(dA:dT) nucleic acid mimics in PMs also downregulated

NDR1 expression (Supplementary Fig. 1c). Viral infection induces the production of type I IFN which subsequently binds to heterodimers of IFNα receptor 1 (IFNαR1) and IFNαR2 and triggers JAK-STAT signaling, thereby amplifying antiviral ISGs expression[2]. We next treated mouse PMs with recombinant mouse IFNβ or Thp1 cells with recombinant human IFNα. As shown in Fig. 1d, e and Supplementary Fig. 1d, e, IFNβ or IFNα stimulation significantly decreased NDR1 expression at both the mRNA and protein levels. Notably, the VSV infection-induced downregulation of NDR1 expression was abolished in IFNαR- or STAT1-deficient macrophages (Fig. 1f, g and Supplementary Fig. 1f, g), thus indicating that viral infection inhibits NDR1 expression depending on STAT1 pathway.

To our interesting, we found a putative STAT1 binding site (5′-TTNCNNNAA-3′: nt −553 to −545)[14] in the human NDR1 promoter region. Moreover, ChIP assay showed that STAT1 bound to NDR1 promoter region in an IFNα stimulation-dependent manner (Supplementary Fig. 1h), which indicating that STAT1 may act as a negative-transcriptional regulator for NDR1 transcription during IFNα stimulation. Histone modifications play important roles in the control of gene transcription. H3K4 trimethylation (H3K4me3) is a marker of transcriptional activation[15], while high levels of H3K27 trimethylation (H3K27me3) in the coding region generally correlate with transcription repression[16]. By ChIP assay, we found that H3K27me3 was upregulated in the promoter of NDR1 gene in Thp1 cells after VSV infection or IFNα stimulation, whereas H3K4me3 remained constant throughout the infection with VSV or IFNα stimulation (Fig. 1h, i). Together, these data demonstrate that viral infection downregulates NDR1 expression via an IFN signaling-dependent mechanism and may also involve epigenetic modification of histone H3, thus indicating that NDR1 may play a crucial role in antiviral innate immunity.

### NDR1 promotes the antiviral immune response in macrophages.
We analyzed the expression of NDR1 in primary mouse immune cells via real-time PCR and found that NDR1 expression was much higher in macrophages than in other immune cell types (Supplementary Fig. 2a). To further determine the function of NDR1 in innate immunity, we obtained PMs from a pair of WT and NDR1-deficient $(NDR1^{-/-})$ littermate mice to assess the effects of NDR1 on viral infection-induced type I IFN, proinflammatory cytokines and antiviral ISGs production. As shown in Fig. 2a, b, NDR1 deficiency significantly inhibited the VSV infection-induced expression of IFNβ, IL6, and TNFα at both the protein and mRNA levels in macrophages. Decreased ISGs induction was also observed in $NDR1^{-/-}$ macrophages after VSV infection (Fig. 2b). NDR1 silencing decreased the VSV-induced increases in IFNβ, proinflammatory cytokines and ISGs in mouse primary macrophages (Supplementary Fig. 2b, c) and human PBMCs (Supplementary Fig. 2d). Furthermore, intracellular poly (I:C)-induced type I IFN, proinflammatory cytokines production (Supplementary Fig. 2e) and antiviral ISGs levels (Supplementary Fig. 2f) were much lower in $NDR1^{-/-}$ macrophages than in WT macrophages. In contrast, the RAW264.7 cell line stably over-expressing NDR1 or its kinase-inactive mutants K118A, K281A/T444A (AA)[17] showed much higher levels of type I IFN, proinflammatory cytokines and antiviral ISGs after VSV infection than did the control transfectants (Supplementary Fig. 2g), a result suggesting that NDR1 promotes virus-induced cytokine production in a kinase-independent manner. The replication of VSV expressing eGFP in RAW264.7 cells and mouse PMs was determined through fluorescence-activated cell sorting analysis and the 50% tissue culture infective dose (TCID50) assay. Whereas overexpression of NDR1 and its kinase-inactive mutants inhibited

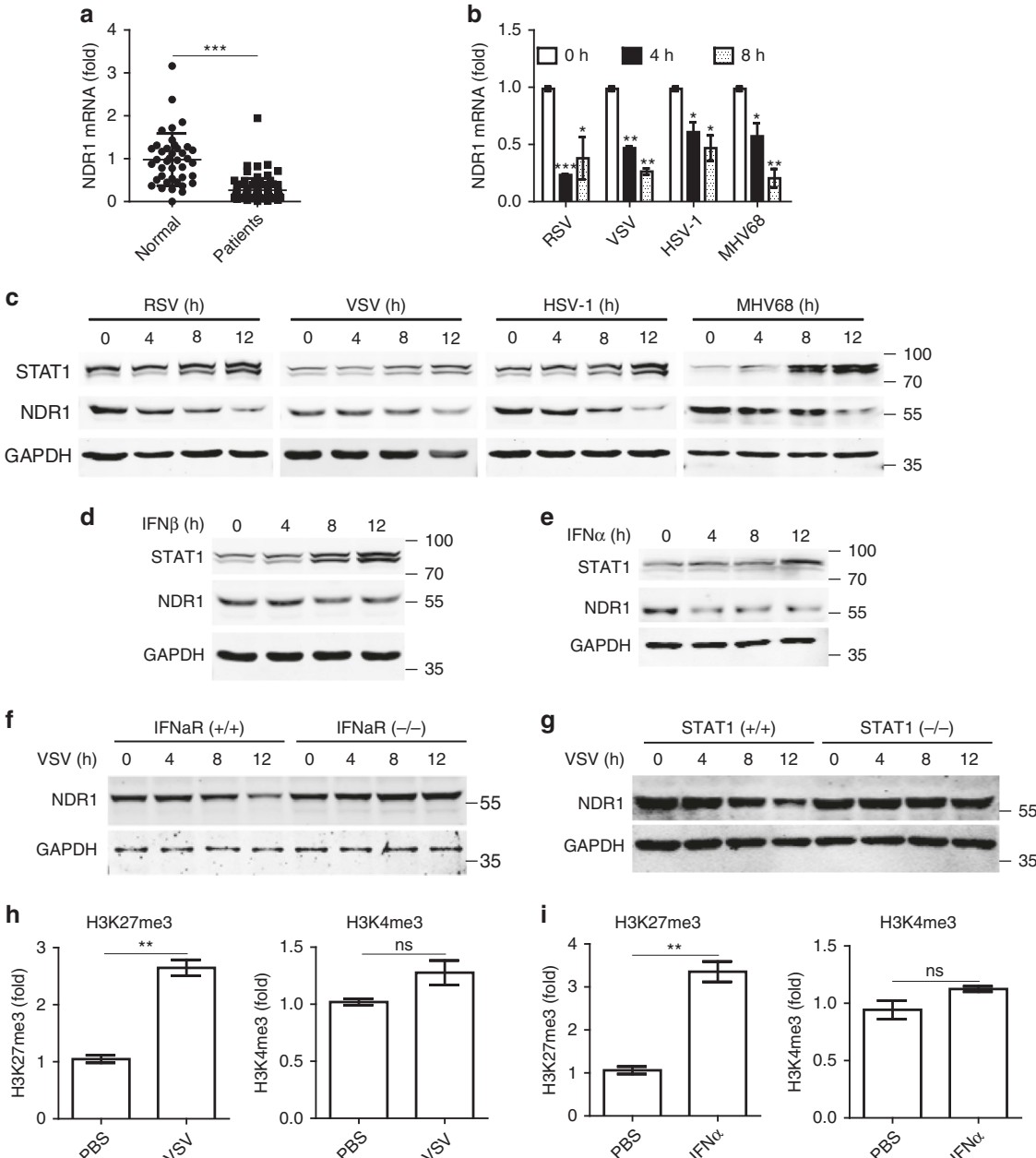

**Fig. 1** Viral infection downregulates NDR1 expression in a type I IFN pathway-dependent manner. **a** Real-time PCR analysis of NDR1 mRNA in the peripheral blood from patients infected with RSV ($n = 77$) and healthy control ($n = 40$). **b**, **c** Real-time PCR (**b**) and immunoblot analysis (**c**) of NDR1 expression in peritoneal macrophages (PMs) infected with RSV, VSV, HSV, or MHV68. **d**, **e** Immunoblot analysis of NDR1 expression in PMs treated with IFNβ (100 IU ml$^{-1}$) (**d**), or with IFNα (100 IU ml$^{-1}$) (**e**). **f**, **g** Immunoblot analysis of NDR1 expression in lysates of $IFN\alpha R^{+/+}$ and $IFN\alpha R^{-/-}$ (**f**) or $STAT1^{+/+}$ and $STAT1^{-/-}$ (**g**) mouse immortalized bone marrow-derived macrophages infected with VSV. **h**, **i** ChIP-qPCR analysis of the histone modification in NDR1 promoter with H3K27me3 and H3K4me3 antibodies in lysates of Thp1 cells treated with VSV for 8 h (**h**) or with IFNα (100 IU ml$^{-1}$) for 4 h (**i**). Data are mean ± SD and are representative of three independent experiments. Student's $t$ test was used for statistical calculation. *$p < 0.05$, **$p < 0.01$, ***$p < 0.001$. See also Supplementary Fig. 1

VSV replication in stable RAW264.7 transfectants (Supplementary Fig. 2h), NDR1 deficiency rendered the cells susceptible to VSV infection (Fig. 2c). In addition, we infected PMs from WT and $NDR1^{-/-}$ mice with RNA viruses (H1N1 and RSV) and DNA viruses (HSV and MHV68) and observed lower production of IFNβ, IL6, TNFα, and ISGs in $NDR1^{-/-}$ macrophages vs. WT macrophages (Fig. 2d). WT and $NDR1^{-/-}$ PMs were then infected with HSV-eGFP to quantify and visualize HSV replication. $NDR1^{-/-}$ macrophages showed a significantly increased percentage of HSV-eGFP$^+$ cells compared with WT macrophages

(Supplementary Fig. 2i). Collectively, these data suggest that NDR1 promotes the cellular antiviral immune response against both RNA and DNA viral infections. The efficiency of NDR1 knockdown, knockout, and overexpression was confirmed by immunoblotting (Supplementary Fig. 2j).

**NDR1 protects mice from RNA and DNA viral infections**. We next evaluated the importance of NDR1 in mediating host defense against viral infection in vivo. NDR1 knockout ($NDR1^{-/-}$) and

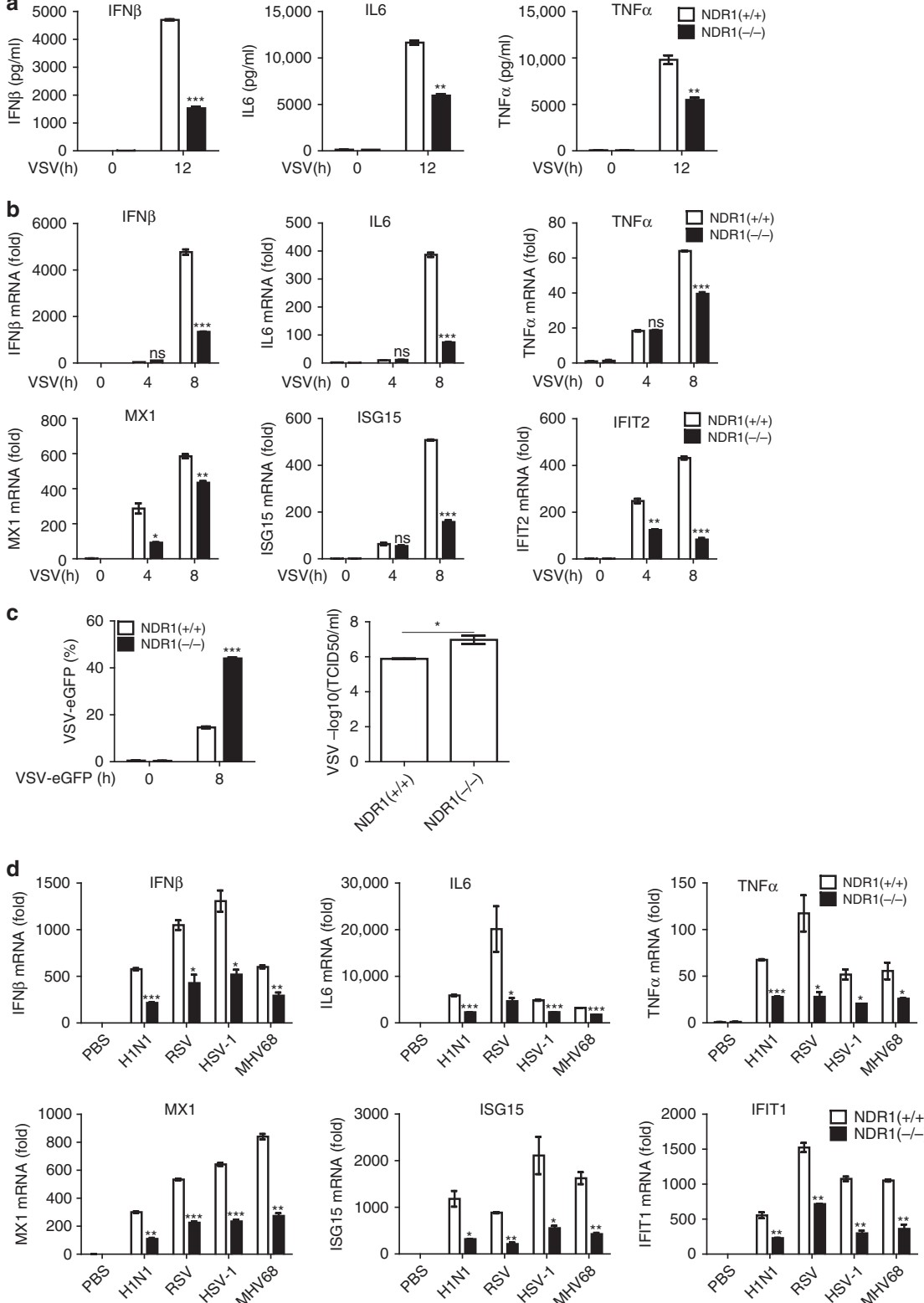

**Fig. 2** NDR1 is required for viral infection-induced IFNβ, proinflammatory cytokines and antiviral ISGs production in macrophages. **a** Enzyme-linked immunosorbent assay (ELISA) of IFNβ, IL6, and TNFα protein in WT and *NDR1*⁻/⁻ PMs infected with VSV. **b** Real-time PCR analysis of IFNβ, IL6, TNFα, and ISGs mRNA in WT and *NDR1*⁻/⁻ PMs infected with VSV. **c** Fluorescence-activated cell sorting (FACS) and TCID50 (50% tissue culture infective dose) assessing the proliferation level of VSV. WT and *NDR1*⁻/⁻ PMs were infected with VSV-eGFP. The fusion protein fluorophore was analyzed by FACS. Then, fresh medium with the VSV-eGFP infection was serially diluted on the monolayer of HEK293 cells for 3–7 days and TCID50 was measured. **d** Real-time PCR analysis of IFNβ, IL6, TNFα, and ISGs mRNA in PMs infected with H1N1, RSV, HSV, and MHV68 for 8 h. Data are mean ± SD and are representative of three independent experiments. Student's *t* test was used for statistical calculation. \*p < 0.05, \*\*p < 0.01, \*\*\*p < 0.001. See also Supplementary Fig. 2

control littermate ($NDR1^{+/+}$) mice were intraperitoneally infected with a sublethal dose of VSV. As shown in Fig. 3a and Supplementary Fig. 3a, $NDR1^{-/-}$ mice exhibited lower levels of IFNβ, IL6, and TNFα in their serum and PMs than WT mice in response to VSV infection. Increased VSV titers with decreased cytokine and ISGs induction were observed in the livers, lungs, and spleens of $NDR1^{-/-}$ mice (Fig. 3b and Supplementary Fig. 3b−d). Furthermore, histopathology revealed much more severe edema, alveolar hemorrhaging, alveolar wall thickening, and neutrophil infiltration in the lungs of $NDR1^{-/-}$ mice compared with WT mice after viral infection (Fig. 3c). Analysis of the liver parenchyma revealed a richer portal inflammatory infiltrate in $NDR1^{-/-}$ mice compared with WT mice, which consisted of mixed eosinophil and monocyte cells (Fig. 3c). The $NDR1^{-/-}$ mice all died within 2 days after infection with a lethal dose of VSV, whereas only 30% of the WT mice succumbed to this infection (Fig. 3d). We administered HSV intravenously to WT and $NDR1^{-/-}$ mice, and monitored the survival of these mice. We observed 80% mortality in $NDR1^{-/-}$ mice 10 days after infection, whereas the mortality of WT mice was only 40% at this time point (Fig. 3e). Similarly, the observed copy numbers of HSV genomic DNA were significantly higher in the brains of $NDR1^{-/-}$ mice than in their WT counterparts (Fig. 3f). In agreement with the survival results, the concentrations of IFNβ and the proinflammatory cytokines IL6 and TNFα in the serum were lower in $NDR1^{-/-}$ mice compared with WT mice after HSV treatment (Fig. 3g). Together, these in vivo data indicate that NDR1 is an important positive regulator of antiviral immune responses against both RNA and DNA viruses.

**NDR1 upregulates STAT1 protein expression.** VSV infection induces type I IFN production by triggering the RNA sensor RIG-I and downstream kinases such as TBK1 and TAK1, which mediate activation of the IRF, NF-κB, and MAPK pathways[18]. Type I IFN binds to IFN receptors and activates the JAK-STAT pathway, thus inducing expression of ISGs, which play an important role in antiviral immunity and positively regulate the RIG-I pathway in feedback[19]. As shown in Fig. 4a and Supplementary Fig. 4a, NDR1 knockout or knockdown slightly inhibited the VSV-induced the activation of the NF-κB, MAPK, IRF3 pathways, and the expression of RIG-I. However, NDR1 knockout or knockdown significantly decreased STAT1 expression in the resting state and inhibited VSV-induced STAT1 phosphorylation (Fig. 4a and Supplementary Fig. 4a). Moreover, overexpression of NDR1 or its kinase-inactive mutants increased STAT1 expression and phosphorylation in RAW264.7 stable cell lines after VSV infection while slightly promoted the activation of the NF-κB and IRF3 pathways and the expression of RIG-I at the 8 h after VSV infection (Fig. 4b). In addition, NDR1 and its kinase-inactive mutants promoted RIG-I-N (a mutant containing the N-terminal CARD domains of RIG-I that acts as a constitutively active RIG-I mutant)-induced IFN-sensitive response element (ISRE) reporter activation (Supplementary Fig. 4b) but not IRF3 reporter activation (Supplementary Fig. 4c). We further tested the roles of NDR1 in the regulation of the type I IFN pathway by using the CRISPR-Cas9-mediated genome editing technique in HEK293 cells to deplete NDR1 expression. NDR1 deficiency in HEK293 cells prevented recombinant human IFNα-induced ISRE reporter activation, which reverted to levels comparable to those of the WT group after overexpression of NDR1 or its kinase-inactive mutants (Supplementary Fig. 4d), a result suggesting that NDR1 promotes the activation of the type I IFN pathway, but not the IRF3 pathway. Together, these results suggest that NDR1 has no direct effect on VSV induced-IRF3 activation, but enhances STAT1 expression and promotes ISGs

induction including RIG-I, which further facilitates the activation of the NF-κB, IRF3 (Fig. 4b) and the production of type I IFN and inflammatory cytokines (Supplementary Fig. 2g) at the 8 h after VSV infection.

Considering the effect of NDR1 on ISGs induction is conserved in mouse and human cells, we next tested whether NDR1 enhanced STAT1 expression in human cells. We found that NDR1 knockout in HEK293 cells (Fig. 4c) and knockdown in human PBMCs (Supplementary Fig. 4e) and Thp1 cells (Supplementary Fig. 4f) inhibited the expression of STAT1, while overexpression of NDR1 in Thp1 cells significantly enhanced the expression of STAT1 (Supplementary Fig. 4g), indicating that the promoting effect of NDR1 on STAT1 expression is conserved between mice and humans. In addition, NDR1 deficiency dampened STAT1 expression with or without RNA or DNA virus infections in various tissues including lung, liver, spleen, and brain (Supplementary Fig. 4h). To further validate the effect of NDR1 on STAT1 expression, we reexpressed NDR1 or its kinase-inactive mutants in $NDR1^{-/-}$ HEK293 cells and macrophages. As shown in Fig. 4c, d, the overexpression of NDR1 or its kinase-inactive mutants in $NDR1^{-/-}$ HEK293 cells and macrophages restored STAT1 expression to the same level as that in control HEK293 cells and WT macrophages, respectively. We also measured the expression of other key molecules involved in the JAK-STAT pathway, including STAT2, STAT3, STAT5, STAT6, and JAK1, in WT and $NDR1^{-/-}$ macrophages. Either NDR1 knockout or knockdown affected the expression of these proteins in macrophages (Fig. 4e, f). Accordingly, the basal protein levels of these molecules were also comparable in RAW264.7 cells overexpressing NDR1 and control transfectants (Fig. 4g). Together, these data demonstrate that NDR1 specifically upregulates the expression of STAT1 and that this upregulation is independent of its kinase activity.

**NDR1 promotes IFN pathway activation and ISGs induction.** It is well established that dimers of phosphorylated STAT1 and STAT2 bind to ISREs and consequently induce antiviral ISGs production after IFN stimulation[19, 20]. Thus, we next investigated the effects of NDR1 on IFN signaling pathways. As shown in Supplementary Fig. 5a, overexpression of NDR1 or its kinase-inactive mutants enhanced IFNβ-induced MX1, ISG15 and IFIT1 expression in RAW264.7 stable cell lines compared with control transfectants. In contrast, compared with WT macrophages, $NDR1^{-/-}$ macrophages exhibited much lower mRNA levels of MX1, ISG15, and IFIT1 after IFNβ treatment (Supplementary Fig. 5b). Notably, reexpression of NDR1 or its kinase-inactive mutants in $NDR1^{-/-}$ macrophages restored IFNβ-induced ISGs expression to a level comparable to that observed in WT macrophages (Fig. 5a). These data further confirm that NDR1 promotes type I IFN pathway signaling in a kinase-independent manner.

To further explore whether NDR1 promotes antiviral innate immune responses through enhancing the type I IFN pathway, we transfected control siRNA or NDR1-specific siRNA into IFNαR$^{+/+}$ and IFNαR$^{-/-}$ macrophages and investigated the effects of NDR1 on the antiviral immune response. Silencing of NDR1 impaired VSV-induced type I IFN and IL6 expression in IFNαR$^{+/+}$ macrophages, but did not affect the expression of these cytokines in IFNαR$^{-/-}$ macrophages (Supplementary Fig. 5c). In addition, silencing of NDR1 promoted viral replication in IFNαR$^{+/+}$ macrophages, but not in IFNαR$^{-/-}$ macrophages (Fig. 5b). Moreover, the inhibitory effects of NDR1 silencing on viral-induced ISGs expression and viral replication were abolished in STAT1-deficient ($STAT1^{-/-}$) macrophages (Fig. 5c, d and Supplementary Fig. 5d). These data demonstrate that NDR1

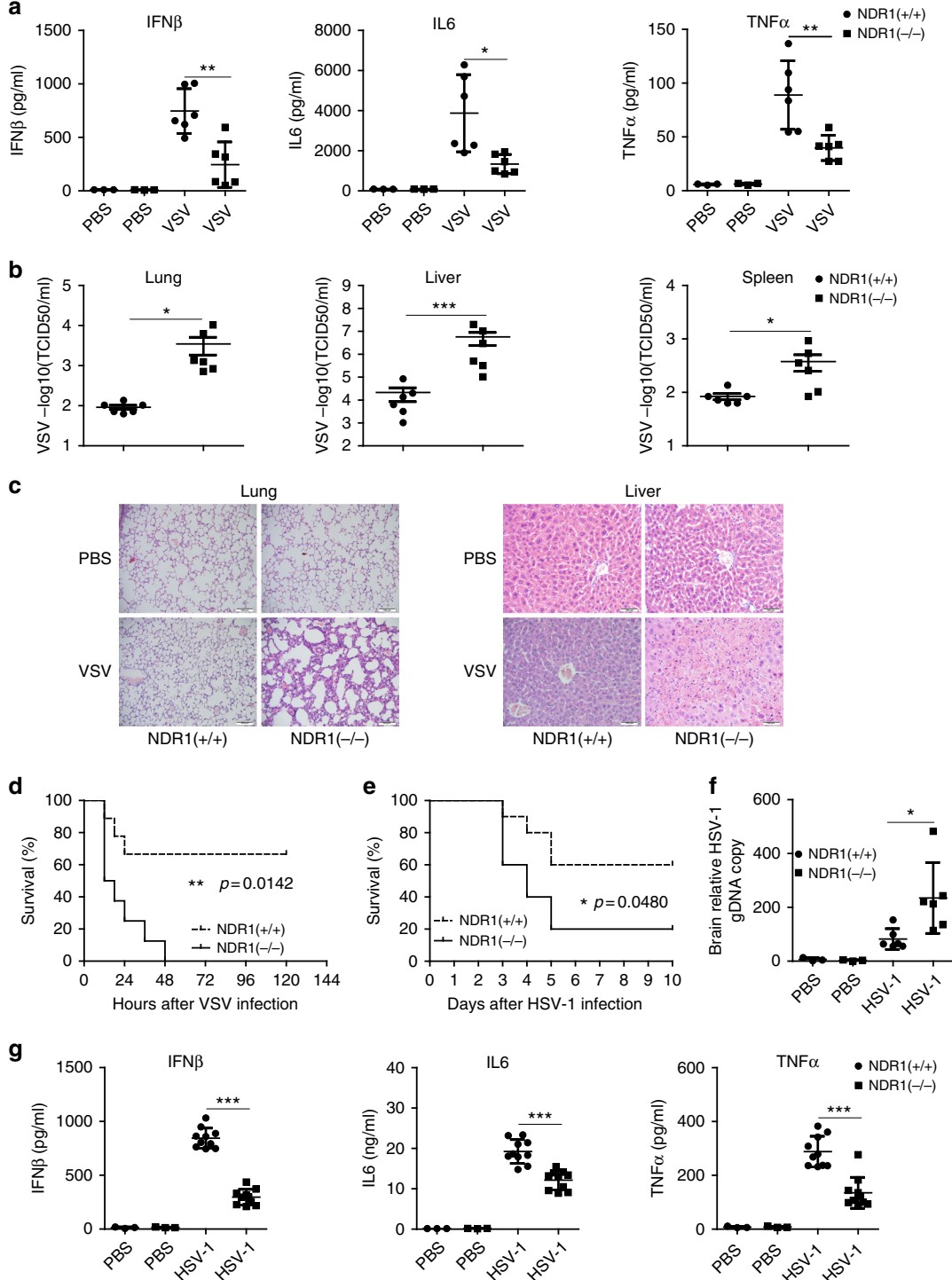

**Fig. 3** NDR1 is required for antiviral innate immune response in vivo. **a** Eight-week-old male *NDR1+/+* and *NDR1−/−* mice (*n* = 6 for each group) were intraperitoneally injected with VSV (1 × 10[7] pfu g[−1]); 12 h later, ELISA of IFNβ, IL6, and TNFα in serum was performed. All mice were housed in SPF conditions. **b** VSV loads in organs were determined by TCID 50 assay from *NDR1+/+* and *NDR1−/−* mice treated as in **a**. **c** Hematoxylin and eosin staining of liver and lung sections from mice in **a**. Scale bar, 100 μm (left) or 50 μm (right). **d** Survival curve for 8-week-old male *NDR1+/+* and *NDR1−/−* mice challenged with VSV (1 × 10[8] pfu g[−1], i.v.) (*n* = 9 per group; Wilcoxon test). All mice were housed in SPF conditions. **e** Survival of 8-week-old male *NDR1+/+* and *NDR1−/−* mice intravenously injected with HSV (1 × 10[6] pfu per mouse) (*n* = 10 per group; Wilcoxon test). All mice were housed in SPF conditions. **f** Real-time PCR analysis of HSV genomic DNA in brains of *NDR1+/+* and *NDR1−/−* mice (*n* = 6 per group) infected with HSV as in **e** for 4 days. **g** ELISA of IFNβ, IL6, and TNFα in serum of *NDR1+/+* and *NDR1−/−* mice (*n* = 10 per group) infected with HSV as in **e** for 6 h. Data are mean ± SD and are representative of three independent experiments. Student's *t* test was used for statistical calculation. *$p < 0.05$, **$p < 0.01$, ***$p < 0.001$. See also Supplementary Fig. 3

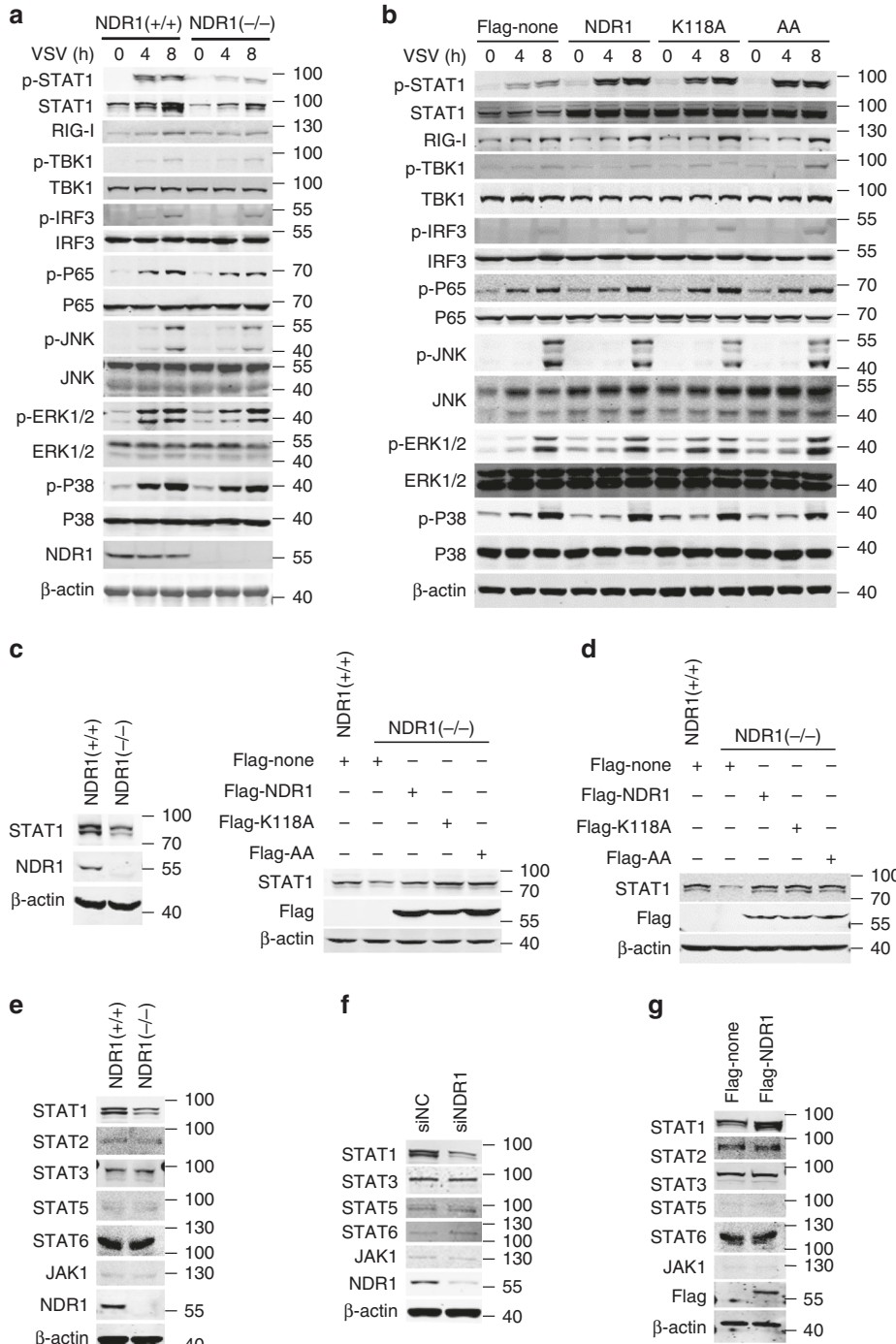

**Fig. 4** NDR1 enhances STAT1 expression in a kinase-independent manner. **a** Immunoblot analysis of total and phosphorylated (p-) STAT1, TBK1, IRF3, P65, ERK1/2, JNK1/2, and P38 in lysates of WT and *NDR1*$^{-/-}$ PMs infected with VSV. **b** Immunoblot analysis of total and phosphorylated (p-) proteins as in **a**, in lysates of RAW264.7 cells stably overexpressing NDR1 or its kinase-inactive mutants, infected with VSV. **c**, **d** Immunoblot analysis of the expression of STAT1 in *NDR1*$^{-/-}$ HEK293 cells (**c**) and PMs (**d**), transfected with vectors expressing NDR1 or its kinase-inactive mutants. **e−g** Immunoblot analysis of the indicated proteins in lysates from the WT and *NDR1*$^{-/-}$ PMs (**e**), the PMs treated with NDR1-specific or scrambled siRNA (**f**), or RAW264.7 cells stably expressing NDR1 (**g**). Data are representative of three independent experiments. See also Supplementary Fig. 4

promotes an antiviral immune response through targeting of STAT1.

Type II IFN is crucial in the cellular response to LM bacteria and acts by triggering homodimerization of phosphorylated STAT1; the STAT1 homodimer then induces antibacterial ISGs expression by binding to the IFNγ activation sequence (GAS)[14]. To determine whether NDR1 plays a critical role in the type II IFN pathway and the antibacterial immune response, we

measured NDR1 levels in PMs in response to LM infection. Both NDR1 mRNA and protein levels were decreased in PMs infected with LM (Supplementary Fig. 5e, f). Furthermore, NDR1 deficiency significantly inhibited LM infection- and IFNγ-induced antibacterial ISGs expression (Supplementary Fig. 5g, h). Notably, mice exhibited higher mortality than WT mice in response to LM infection (Supplementary Fig. 5i). Collectively, these data indicate that NDR1 promotes both type I and type II

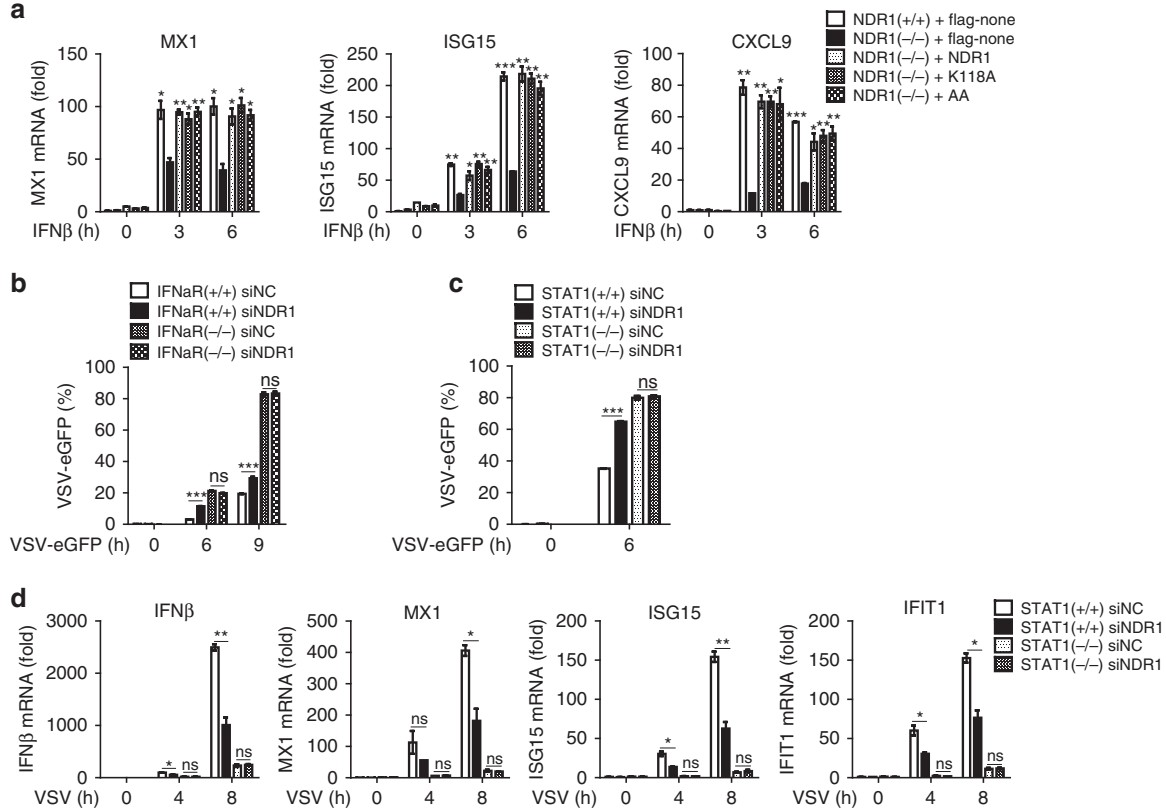

**Fig. 5** NDR1 positively regulates antiviral immune response via promoting STAT1 expression. **a** Real-time PCR analysis of ISGs mRNA in WT or $NDR1^{-/-}$ PMs transfected with vectors expressing NDR1 or its kinase-inactive mutants, followed by stimulation with IFNβ (100 IU ml$^{-1}$). **b**, **c** FACS analysis of immortalized BMDMs obtained from WT and IFNα receptor-deficient mice (**b**) or from WT and $STAT1^{-/-}$ mice (**c**), treated with NDR1-specific or scrambled siRNA for 48 h, followed by VSV-eGFP infection. **d** Real-time PCR analysis of IFNβ and ISGs mRNA in BMDMs as in **c**. Data are mean ± SD and are representative of three independent experiments. Student's $t$ test was used for statistical calculation. *$p < 0.05$, **$p < 0.01$, ***$p < 0.001$. See also Supplementary Fig. 5

IFN pathways and plays an essential role in both the antiviral and antibacterial immune responses.

**NDR1 facilitates STAT1 translation.** We next sought to determine the mechanism by which NDR1 enhances STAT1 expression. We tested the effect of NDR1 on STAT1 transcription via quantitative real-time PCR and observed that NDR1 knockout did not alter the level of STAT1 mRNA in macrophages with or without VSV infection (Supplementary Fig. 6a), thus suggesting that NDR1 has no effect on STAT1 transcription. Subsequently, we examined the effect of NDR1 on STAT1 protein degradation in macrophages after pretreatment with either the proteasome inhibitor MG132 or the autophagy inhibitor chloroquine (CQ). As shown in Supplementary Fig. 6b, c, MG132 or CQ pretreatment did not abrogate the impairment of STAT1 expression in $NDR1^{-/-}$ macrophages. Moreover, the cycloheximide chase assay performed in macrophages showed that the half-life of the endogenous STAT1 protein was comparable in WT and $NDR1^{-/-}$ macrophages (Supplementary Fig. 6d), thus indicating that NDR1 does not regulate STAT1 protein stability.

Given that NDR1 affected neither the transcription of STAT1 nor its protein stability, we next tested whether NDR1 regulates STAT1 translation. We used metabolic labeling with biotin to identify newly synthesized proteins[21]. Whereas, overexpression of NDR1 had no effect on the overall levels of newly synthesized proteins in RAW264.7 cells, it significantly increased the amount of newly synthesized STAT1 protein (Fig. 6a). Accordingly, NDR1 knockout or knockdown markedly diminished the amount

of newly synthesized STAT1 protein in PMs (Fig. 6b, c) and BMDMs (Supplementary Fig. 6e). In addition, a decrease in the level of newly synthesized STAT1 protein was also observed in $NDR1^{-/-}$ HEK293 cells compared with WT HEK293 control cells (Supplementary Fig. 6f). Together, these data indicate that NDR1 promotes STAT1 translation.

**Nuclear NDR1 binds the intergenic region of miR146a.** NDR1 was originally defined as a nuclear protein containing a non-consensus nuclear localization signal (NLS) in its catalytic domain insert[22]; however, recent findings provide strong evidence that NDR1 is also a cytoplasmic kinase[23]. In accordance with these findings, our immunoblot assays and confocal experiments showed that NDR1 was located in both the nuclei and cytoplasm of macrophages with or without VSV infection (Supplementary Fig. 6g, h). As expected, deletion of NLS amino acids 265–276 of NDR1 (ΔNLS) inhibited its translocation to the nucleus (Fig. 6d), and abolished its effects on STAT1 expression in RAW264.7 stable transfectants (Fig. 6e) and HEK293 cells (Supplementary Fig. 6i). Accordingly, the decreased expression of STAT1 in $NDR1^{-/-}$ macrophages was restored by overexpression of NDR1 or its kinase-inactive mutant K118A but not by overexpression of the NDR1 ΔNLS mutant (Supplementary Fig. 6j). Consequently, overexpression of the NDR1 ΔNLS mutant did not enhance IFN-β-induced ISGs expression (Supplementary Fig. 6k) or inhibit VSV replication in RAW264.7 stable cell lines (Fig. 6f). Together, these results suggest that nuclear NDR1 promotes STAT1 translation and functions as an antiviral molecule.

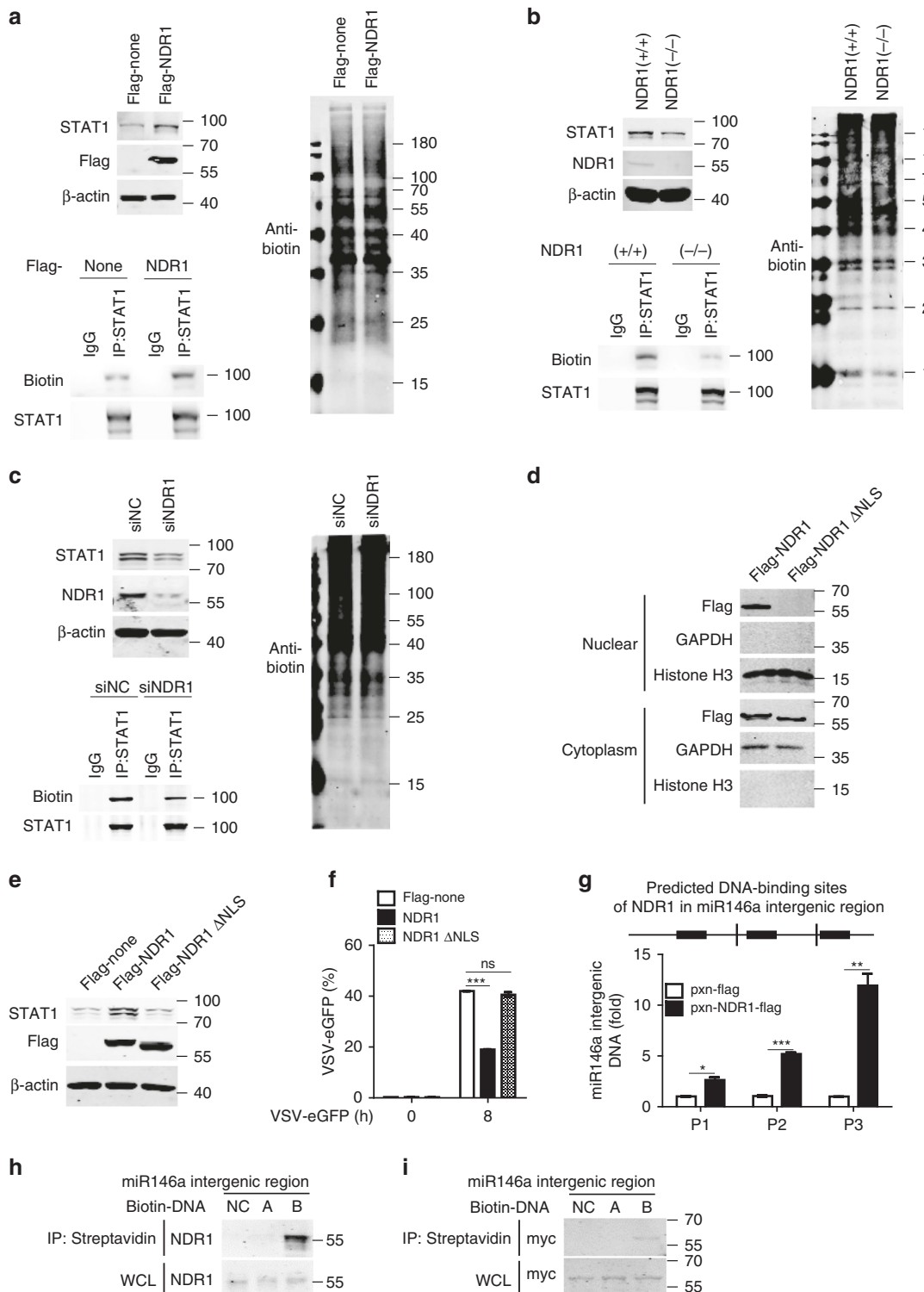

**Fig. 6** Nuclear NDR1 promotes STAT1 translation. **a**−**c** RAW264.7 cells stably expressing flag-NDR1 or control transfectants (**a**), WT or *NDR1*[−/−] PMs (**b**), NDR1 silencing PMs (**c**), were treated with metabolic labeling reagents (AHA). Subsequently corresponding cell lysates were labeled with biotin for nascent proteins via azide/alkyne reaction, followed by immunoprecipition with anti-STAT1 antibody or nonspecific IgG. **d** Immunoblot analysis of flag-tagged full-length NDR1 or NDR1 ΔNLS mutant expression in cytoplasmic and nuclear fractions from RAW264.7 cells stably expressing flag-NDR1 or flag-NDR1 ΔNLS mutant. **e** Immunoblot analysis of STAT1 expression in RAW264.7 cells stably overexpressing NDR1, NDR1 ΔNLS mutant or empty vector. **f** FACS analysis of RAW264.7 cells as in **e** infected with VSV-eGFP. **g** ChIP-qPCR analysis of the enrichment of NDR1 to miR146a intergenic region with anti-flag M2 magnetic beads in lysates of RAW264.7 cells stably expressing flag-NDR1 or control transfectants. **h** Biotin-labeled DNA was incubated with lysates of PMs for 30 min at 25 °C. Streptavidin sepharose beads were then added and incubated for another 30 min. The beads were washed and then resolved by 2× SDS loading buffer for immunoblotting. **i** Biotin-labeled DNA (miR146a intergenic region or control DNA) and purified myc-NDR1 protein were incubated in DNA pull-down buffer for 30 min at 25 °C. Streptavidin sepharose beads were then added and treated as in **h** for immunoblotting. Data are mean ± SD and are representative of three independent experiments. Student's *t* test was used for statistical calculation. *$p < 0.05$, **$p < 0.01$, ***$p < 0.001$. See also Supplementary Fig. 6

It has been reported that NDR1 binds to nucleotides via amino acids 95–103 in the catalytic domain[24]; thus, on the basis of our finding that nuclear NDR1 enhanced STAT1 translation, we hypothesized that NDR1 might function as a transcriptional regulator. To test this hypothesis, ChIP coupled with high-throughput sequencing (ChIP-seq) was performed in RAW264.7

cells stably overexpressing flag-NDR1; we used anti-flag magnetic beads, because a commercial anti-NDR1 antibody for ChIP assays is currently unavailable. This analysis revealed a total of 477 NDR1-binding peaks, among which 41.09, 11.32, 42.98, and 2.10% were located in intronic regions, exonic regions, intergenic regions, and promoter regions, respectively (Supplementary

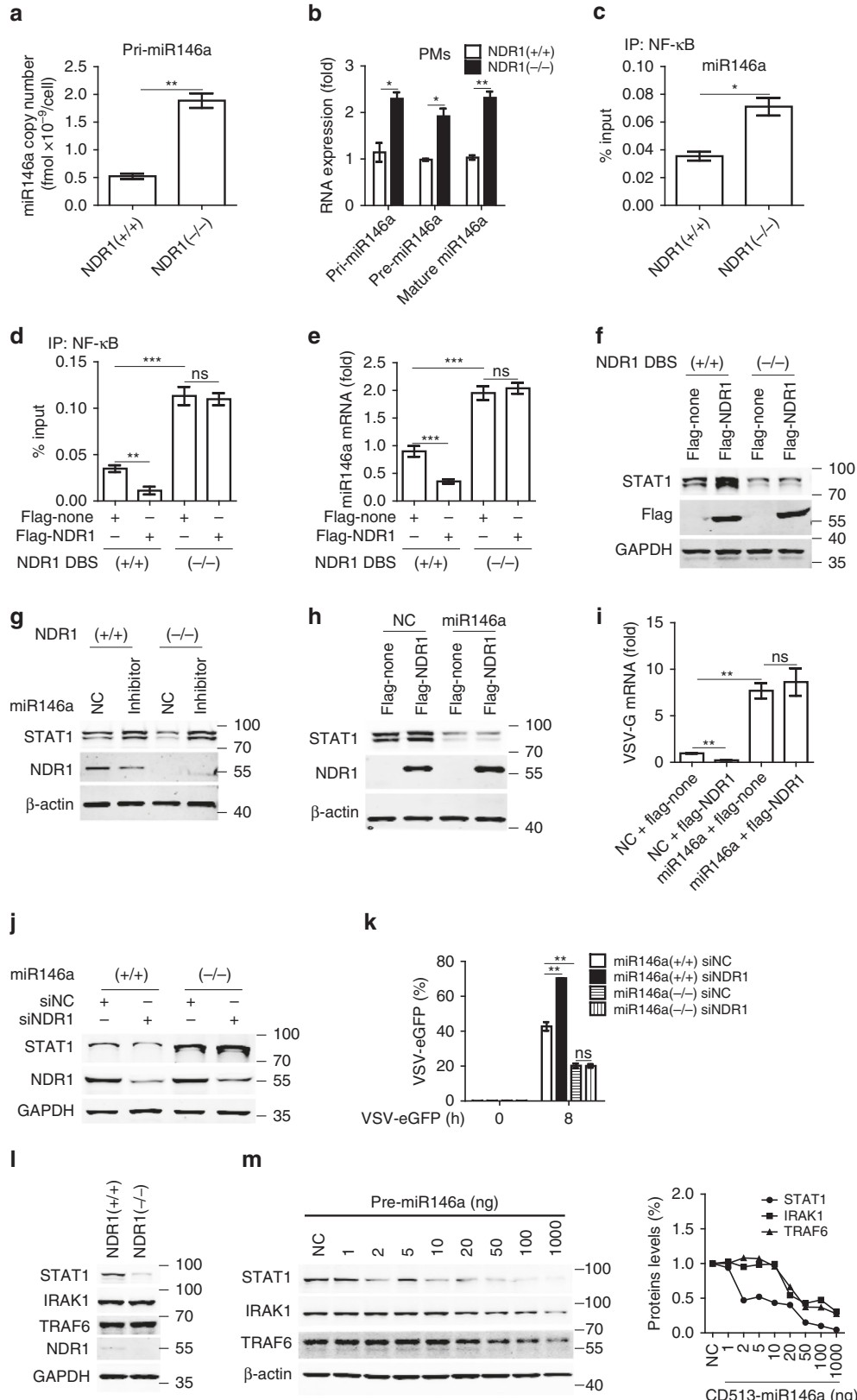

Fig. 6l). Analysis of the NDR1 binding peaks identified the sequence 5′-WCHTBRMAATCG-3′ (W = A or T; H = A, T, or C; B = G, T, or C; R = A or G; and M = A or C) as the consensus motif (Supplementary Fig. 6m).

Notably, among the 477 NDR1-binding peaks, we noted one peak (chr11: 43303594-43303962) in the miR146a intergenic region (Supplementary Fig. 6n). MiR146a has been reported to inhibit STAT1 translation and to impair the IFN-induced anti-HBV immune response in hepatocytes[5]. Thus, we next tested whether the NDR1 promotion of STAT1 translation occurred via targeting of miR146a. According to the NDR1-binding DNA motif that we analyzed, we identified two putative NDR1 binding sites in the miR146a intergenic region (site A: 43303612-43303622 and site B: 43303808-43303818). To validate the ChIP-seq data, we designed three pairs of primers to analyze NDR1 binding sites in the miR146a intergenic region (Supplementary Fig. 6n). As shown in Fig. 6g, we found that NDR1 was more highly enriched at site B compared with site A of the miR146a intergenic region. To further demonstrate that NDR1 binds to the miR146a intergenic region, we chemically synthesized biotin-labeled DNA sequences of these two sites and biotin-labeled negative control DNA (chr11: 43303353-43303383), and then mixed the DNA with cell lysates from primary PMs or HEK293 cells overexpressing flag-tagged NDR1 or its kinase-inactive mutants. As shown in Fig. 6h, endogenous NDR1 was precipitated by only biotin-labeled site B DNA. Accordingly, both WT NDR1 and its kinase-inactive mutants were detected in the precipitate obtained by using biotin-labeled site B DNA (Supplementary Fig. 6o). To verify that NDR1 was able to directly bind to the miR146a intergenic region, the recombinant myc-NDR1 protein was incubated with the biotin-labeled DNA sequence of the miR146a intergenic region, and a DNA pull-down assay was performed. Indeed, we found that the recombinant myc-NDR1 protein was detected in only the precipitate obtained with biotin-labeled site B DNA, but not precipitates obtained with the biotin-labeled negative control or site A DNA (Fig. 6i). Together, these results show that NDR1 directly binds to the miR146a intergenic region (chr11: 43303808-43303818) in a kinase-independent manner.

**NDR1 promotes STAT1 translation via targeting miR146a.**
The 17–26 nucleotide (nt) single-stranded miRNA molecules derive from primary transcripts pri-miRNA, which contains a small number of clustered miRNA units. Following transcription, pri-miRNA is processed into ~ 70 nt stem loop precursors (pre-miRNA) by the Drosha RNAse III nuclease. Those pre-miRNA molecules are transported to cytoplasm and possessed to their final mature form (17–26 nt) by RNAse III nuclease Dicer[25]. We next investigated the effect of NDR1 on the mRNA processing of miR146a in macrophages. At first, levels of miR-146a primary transcript (pri-miR146a) were detected to examine whether NDR1 affects miR-146a transcription. Absolute quantitative real-time PCR assay of pri-miR146a was performed and the standard curve was generated by using pre-miR146a overexpressing plasmid, which consists of the stem−loop structure and more than 200 base pairs of upstream and downstream flanking genomic sequence of pre-miR146a. As shown in Fig. 7a and Supplementary Fig. 7a, the copy number per cell of pri-miR146a was higher in NDR1-deficient immortalized BMDMs ($1.8 \times 10^{-9}$ fmol per cell) than in WT immortalized BMDMs ($0.5 \times 10^{-9}$ fmol per cell) (Fig. 7a), while NDR1 overexpressed RAW264.7 cells showed lower pri-miR146a ($1.6 \times 10^{-10}$ fmol per cell) than the control transfectants ($3.0 \times 10^{-10}$ fmol per cell) (Supplementary Fig. 7a). As expected, NDR1 knockout in mouse PMs or knockdown in human PBMCs increased pri-, pre-, and mature-miR146a expression by relative quantitative real-time PCR (Fig. 7b and Supplementary Fig. 7b). The overexpression of NDR1 in RAW264.7 or Thp1 cells inhibited pri-, pre-, and mature-miR146a expression (Supplementary Fig. 7c). However, the overexpression of NDR1 ΔNLS mutant had no effect on mature miR146a expression in RAW264.7 cells (Supplementary Fig. 7d). In addition, silencing NDR1 enhanced mature miR146a expression in various kinds of cells (Supplementary Fig. 7e). Moreover, NDR1 deficiency significantly promoted the expression of mature miR146a in lung, liver, spleen, and brain in both resting and infectious states (Supplementary Fig. 7f, g). Northern blot assay was also performed to explore the effect of NDR1 on the miR146a processing. Considering that the radiative facility is not available, we used the digoxigenin system to examine the miR146a processing in Thp1 cells. As shown in Supplementary Fig. 7h, only the expression of mature miR146a but not the expression of pri- and pre-miR146a was detectable in Thp1 cell and NDR1 knockdown significantly upregulated the expression of mature miR146a. Taken together, these results suggest that nuclear NDR1 decreases miR146a expression by dampening its transcription, which is conserved between mice and humans.

MiR-146a expression in human hematopoietic cells is known to be transcriptionally induced by NF-κB[4]. Given that NDR1 inhibits miR146a expression, we further explored whether NDR1 decreases NF-κB-mediated miR146a transcription. ChIP assays

---

**Fig. 7** NDR1 binds to miR146a intergenic region to dampen NF-κB-mediated miR146a transcription. **a** Absolute quantification PCR analysis of pri-miR146a expression in WT and *NDR1*$^{-/-}$ immortalized BMDMs. **b** Real-time PCR analysis of pri-, pre-, and mature-miR146a expression in WT and *NDR1*-deficient PMs. TaqMan microRNA assay using specific hydrolysis probes for mature miR146a and U6 was performed to achieve specificity. **c** ChIP-qPCR analysis of NF-κB enrichment in miR146a regulatory sequences with anti-NF-κB antibody in lysates of WT or *NDR1*$^{-/-}$ PMs. **d** ChIP-qPCR analysis of NF-κB enrichment in miR146a regulatory sequences with anti-NF-κB antibody in lysates of NDR1 DNA binding sites deficient (NDR1 DBS$^{-/-}$) and control (NDR1 DBS$^{+/+}$) L929 cells transfected with flag-NDR1 or empty vector. **e** Real-time PCR analysis of mature miR146a expression in NDR1 DBS$^{+/+}$ and NDR1 DBS$^{-/-}$ L929 cells transfected with flag-NDR1 or empty vector. **f** Immunoblot analysis of STAT1 expression in NDR1 DBS$^{+/+}$ and NDR1 DBS$^{-/-}$ L929 cells transfected with flag-NDR1 or empty vector. **g** Immunoblot analysis of STAT1 expression in WT and *NDR1*$^{-/-}$ PMs, transfected with miR146a inhibitor or negative control. **h** Immunoblot analysis of STAT1 in lysates of L929 cells stably overexpressing pre-miR146a or empty vector, transfected with plasmids expressing flag-NDR1 or flag-none. **i** Real-time PCR analysis of VSV-G mRNA in L929 cells stably overexpressing pre-miR146a or empty vector, treated as in **h** and followed by VSV infection for 8 h. **j** Immunoblot analysis of STAT1 in lysates of WT and *miR146a*$^{-/-}$ L929 cells treated with NDR1-specific or scrambled siRNA for 48 h. **k** FACS analysis of virus replication in WT and *miR146a*$^{-/-}$ L929 cells treated as in **j**, followed by VSV-eGFP infection. **l** Immunoblot analysis of STAT1/IRAK1/TRAF6 expression in the lysates of WT and *NDR1*$^{-/-}$ PMs. **m** Immunoblot analysis of STAT1/IRAK1/TRAF6 expression in the lysate of HEK293 cells transfected with empty vector or increasing concentrations of miR146a constructs (left). Proteins densitometry was presented relative to β-actin and normalized with their densitometry on the resting state (right). Data are mean ± SD and are representative of three independent experiments. Student's *t* test was used for statistical calculation. *$p < 0.05$, **$p < 0.01$, ***$p < 0.001$. See also Supplementary Fig. 7

showed that NDR1 overexpression significantly inhibited the enrichment of NF-κB to miR146a promoter[4] (Supplementary Fig. 7i), whereas NDR1 silencing facilitated NF-κB binding to the miR146a promoter in Thp1 cells (Supplementary Fig. 7j). Jayeeta et al. have reported that NF-κB binds to regulatory sequences (−10,680 to −11,120) in the upstream of miR-146a in mouse striatal cells to induce the transcription of miR146a[26]. Considering that the inhibitory effect of NDR1 on miR146a transcription is conserved in humans and mice, we next explored whether NDR1 dampened NF-κB binding to regulatory sequences of miR146a in mouse cells. As shown in Fig. 7c and Supplementary Fig. 7k, NDR1 deficiency significantly facilitated NF-κB binding to the miR146a regulatory sequences in mice PMs (Fig. 7c), whereas overexpressing NDR1 attenuated the enrichment of NF-κB at the miR146a regulatory sequences (Supplementary Fig. 7k). Because NF-κB is critical for the transcription of IL6, TNFα and IL1β[27], we also investigated whether NDR1 affected NF-κB binding to the promoters of these genes. As shown in Supplementary Fig. 7l, m, NDR1 did not affect the enrichment of NF-κB at the promoters of these genes, thus possibly explaining why we observed no effect of NDR1 on these downstream mediators at 4 h after VSV infection (Fig. 2b and Supplementary Fig. 2d).

We further investigated whether NDR1 decreases NF-κB-mediated miR146a transcription depending on the binding of NDR1 to the intergenic region of miR146a. The DNA binding site of NDR1 in miR146a intergenic region was knocked out (NDR1 DBS$^{-/-}$) in L929 cells using CRISPR-Cas9 system (Supplementary Fig. 7n). As shown in Fig. 7d, overexpressing NDR1 attenuated the enrichment of NF-κB at the miR146a regulatory sequences in control L929 cells (NDR1 DBS$^{+/+}$) but this effect was abolished in NDR1 DBS$^{-/-}$ L929 cells. Moreover, the effects of NDR1 on the expression of miR146a, STAT1, and viral-induced ISGs were repealed in NDR1 DBS$^{-/-}$ L929 cells (Fig. 7e, f and Supplementary Fig. 7o). Taken together, these results indicate that NDR1 dampens NF-κB-mediated miR146a transcription in an miR146a intergenic region-binding dependent manner.

To explore whether NDR1 upregulates STAT1 translation by targeting miR146a, we transiently transfected a miR146a inhibitor into NDR1 knockout and knockdown macrophages, or miR146a mimics into RAW264.7 cells stably overexpressing NDR1. As shown in Fig. 7g and Supplementary Fig. 7p, knockout or silencing of NDR1 impaired STAT1 expression but did not affect STAT1 expression in miR146a inhibitor-transfected macrophages. In addition, overexpression of miR146a mimics not only inhibited STAT1 expression but also restored the protein level of STAT1 in stable transfectants overexpressing NDR1, to a level comparable to that observed in the control transfectants (Supplementary Fig. 7q). In line with results from previous studies[5], inhibition of miR146a expression enhanced the protein level of STAT1, IRAK1, and TRAF6 (Supplementary Fig. 7r), while overexpression of miR146a mimics downregulates the expression of these proteins (Supplementary Fig. 7s). We further generated pre-miR146a stably overexpressed murine fibroblast L929 cells to investigate whether the role of NDR1 on STAT1 translation and the antiviral immune response was dependent on miR146a. As shown in the Fig. 7h and Supplementary Fig. 7t, overexpression of NDR1 upregulated STAT1 expression and VSV-induced ISGs expression in control group, but had no effect on STAT1 protein level and ISGs induction in pre-miR146a overexpressed L929 cells. Consistently, the inhibitory effect of NDR1 overexpression on VSV replication was abolished in the pre-miR146a overexpressed L929 cells (Fig. 7i). Meanwhile, miR146a knockout (miR146a$^{-/-}$) L929 cells and HEK293 cells were generated. Silencing NDR1 inhibited STAT1 expression and VSV-induced ISGs expression in miR146a$^{+/+}$ cells but had no

effect on STAT1 protein level and ISGs induction in miR146a$^{-/-}$ L929 cells (Fig. 7j and Supplementary Fig. 7u) and miR146a$^{-/-}$ HEK293 cells (Supplementary Fig. 7v). Furthermore, NDR1 knockdown promoted virus replication in control cells, an effect that also disappeared in miR146a$^{-/-}$ L929 cells (Fig. 7k) and miR146a$^{-/-}$ HEK293 cells (Supplementary Fig. 7w). Together, these data indicate that miR146a is required for NDR1's upregulation of STAT1 expression and stimulation of the antiviral immune response.

IRAK1, TRAF6, and STAT1 have been identified as targets of miR146a[5,7,8,28]. However, NDR1 deficiency attenuated the expression of STAT1, but had no effect on IRAK1 and TRAF6 expression (Fig. 7l). We next explored the sensitivity of STAT1, IRAK1, and TRAF6 to miR146a-mediated inhibition. HEK293 cells were transfected with pre-miR146a overexpressing plasmid, which mimics the natural expression of miR146a in the cells. As shown in Fig. 7m, transfecting low dose (2 ng) of pre-miR146a overexpressing plasmid significantly attenuated STAT1 expression, while at least overexpressing 20 ng of pre-miR146a plasmid could inhibit the expression of IRAK1 and TRAF6. We found that transfecting 2 ng of pre-miR146a overexpressing plasmids resulted in about twofold more miR146a expression in HEK293 cells compared to the control transfectants (Supplementary Fig. 7x). In addition, we observed that NDR1 deficient macrophages expressed only about twofold more miR146a than WT macrophage in the resting state (Fig. 7b). Together, these data indicate that miR146a is delicately regulated by NDR1 at the basal level and STAT1 is more sensitive to miR146a-mediated inhibition than IRAK1 and TRAF6, which may be an underlying mechanism by which NDR1 specifically upregulates STAT1 but does not affect IRAK1 or TRAF6 in the resting state.

**STAT1 binds to miR146a promoter to inhibit its expression.** Coregulation between miRNAs and their target transcription factors is prevalent in living cells[29]. It has been reported that a regulatory feedback loop between STAT1 and miR-155-5p is involved in tumorigenesis[30]. In our model system, STAT1 knockout or knockdown significantly enhanced the expression of miR146a in macrophages (Fig. 8a, b). Thus, we searched the murine miR146a regulatory sequences and found several putative STAT1 binding sites. Multisite ChIP-qPCR analysis showed that STAT1 was enriched in the −10,785 to −10,776 regions of the murine miR146a regulatory sequences (Fig. 8c). Because the STAT1 binding site substantially overlaps the NF-κB binding site in the miR146a regulatory sequences, we hypothesized that STAT1 and NF-κB might competitively bind to the miR146a regulatory sequences. A ChIP assay was performed in WT and STAT1-deficent macrophages with an anti-NF-κB antibody. We found that STAT1 deficiency enhanced NF-κB binding to miR146a regulatory sequences (Fig. 8d). Consistently, silencing of STAT1 resulted in increased NF-κB enrichment at the miR146a promoter in Thp1 cells (Fig. 8e). We next explored whether NDR1 inhibits miR146a expression via STAT1. As shown in Fig. 8f, g, NDR1 silencing enhanced miR146a expression in both WT and STAT1$^{-/-}$ macrophages, thus suggesting that NDR1 directly binds to the miR146 intergenic region and inhibits miR146a expression in a STAT1-independent manner at the basal level. Together, these data indicate that STAT1 and miR146a mutually inhibit each other, such that miR146a downregulates STAT1 translation by targeting the 3′-UTR of STAT1 mRNA, and STAT1 binds to the miR146a promoter and inhibits NF-κB-mediated miR146a transcription.

**NDR1 determines hepatocyte resistance to HCV infection.** Efficient ISGs induction is linked to the successful clearance of

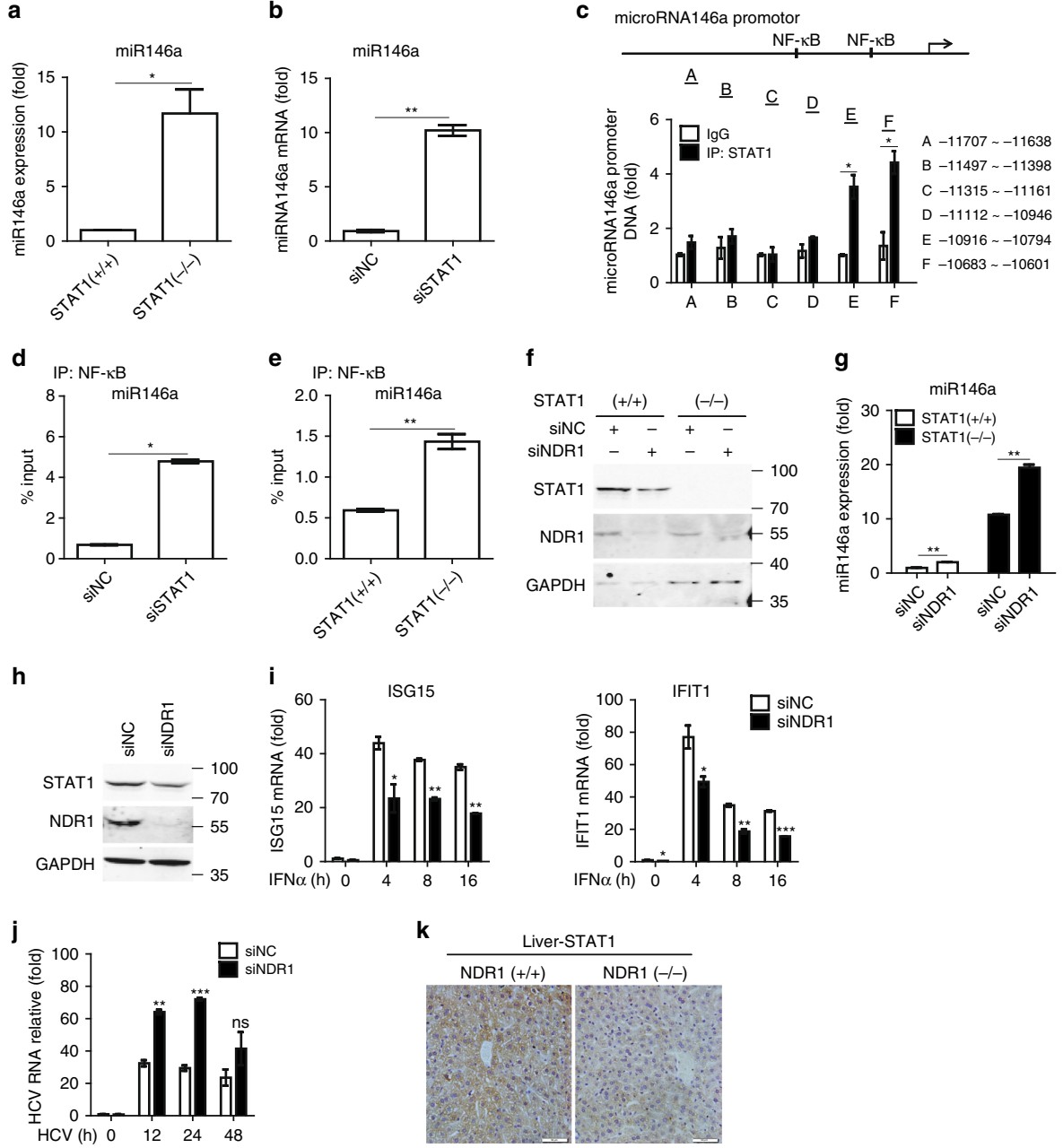

**Fig. 8** STAT1 binds to the miR146a promoter and inhibits miR146a transcription. **a**, **b** Real-time PCR analysis of miR146a expression in *STAT1*[−/−] immortalized BMDMs (**a**) or Thp1 cells treated with STAT1-specific or scrambled siRNA for 48 h (**b**). **c** ChIP-qPCR analysis of the enrichment of STAT1 to six regions (capital letters A–F) of miR146a regulatory sequences with anti-STAT1 antibody in PMs. **d** ChIP-qPCR analysis of the enrichment of NF-κB to miR146a promoter with anti-NF-κB antibody in lysates of Thp1 cells treated with NDR1-specific or scrambled siRNA for 48 h. **e** ChIP-qPCR analysis of the enrichment of NF-κB to miR146a regulatory sequences with anti-NF-κB antibody in lysates of WT and *STAT1*[−/−] immortalized BMDMs. **f** Immunoblot analysis of STAT1 expression in *STAT1*[−/−] immortalized BMDMs treated with NDR1-specific or scrambled siRNA for 48 h. **g** Real-time PCR analysis of miR146a expression in immortalized BMDMs obtained from WT or *STAT1*[−/−] mice treated as in **f**. **h** Immunoblot analysis of STAT1 expression in Huh7 cells treated with NDR1-specific or scrambled siRNA for 48 h. **i** Real-time PCR analysis of ISGs mRNA in Huh7 cells treated as in **h**, followed by stimulation with IFNα (100 IU ml[−1]). **j** Real-time PCR analysis of the HCV genome in Huh7 cells treated as in **h**, followed by infection with HCV JFH1. **k** Representative immunohistochemical staining of liver from WT and NDR1-deficient mice with anti-STAT1 antibody. Data are mean ± SD and are representative of three independent experiments. Student's *t* test was used for statistical calculation. *$p < 0.05$, **$p < 0.01$, ***$p < 0.001$

viruses, including hepatitis C virus (HCV)[31]. Synthesized IFN has been used as a central component of anti-HCV therapy in a clinical setting[32]. Given that NDR1 positively regulates STAT1 expression and facilitates ISGs induction after viral infection, we tested whether NDR1 might prevent HCV infection. As shown in Fig. 8h, NDR1 knockdown significantly inhibited STAT1 expression in Huh7 cells, an immortalized PHH cell line

(PH5CH8) that provides a foundation for in vitro HCV virological studies. As expected, NDR1 knockdown inhibited ISGs induction in Huh7 cells treated with synthesized IFNα (Fig. 8i). In addition, HCV replication was enhanced in NDR1 knockdown Huh7 cells compared with controls (Fig. 8j). Immunohistochemical staining was performed to test the effect of NDR1 on STAT1 expression in vivo. As shown in Fig. 8k, STAT1

expression was markedly decreased in livers of $NDR1^{-/-}$ mice. Together, these data suggest that NDR1 may be involved in the regulation of acute HCV infection and may hold promise in antiviral therapy.

## Discussion

In this study, we demonstrated that NDR1, a positive regulator of ISGs induction, is downregulated after viral infection via STAT1 signaling. The downregulation of NDR1 results in enhanced NF-κB-mediated miR146a transcription, thus perturbing the mutual inhibition between STAT1 and miR146a and ultimately resulting in decreased ISGs induction and an impaired antiviral immune response (Supplementary Fig. 8a, b).

NDR1 is a ubiquitously expressed and highly conserved serine/threonine kinase in the NDR/LATS kinase family. Here, we found that NDR1 is required for defense against viruses and LM invasion in mice. NDR1 knockdown or knockout significantly impaired viral infection-induced type I IFN and ISGs expression, whereas overexpression of NDR1 or its kinase-inactive mutants enhanced viral infection-induced type I IFN and ISGs expression. More viral replication was consistently observed in $NDR1^{-/-}$ macrophages than in WT macrophages. In accordance with the promoting effect of NDR1 on antiviral innate immunity in vitro, NDR1 deficiency also rendered mice more susceptible to viral infection.

After viral infection, viral nucleic acids are sensed by PRRs, including RLRs and cGAS, thereby initiating the activation of type I IFN production pathways[33]. Consequently, IFN binds to its receptor and activates the JAK-STAT pathway, which in turn induces the expression of hundreds of ISGs that play a crucial role in the intracellular antiviral defense program[34]. In the present study, NDR1 has no effect on the viral infection-induced MAPK, NF-κB or IRF3 pathways but specifically affects STAT1 translation. Furthermore, deficiency of IFNαR or STAT1 abolishes the effect of NDR1 on virus clearance, thus indicating that NDR1 promotes the antiviral immune response via targeting STAT1. It has been reported that miR146a binds to the STAT1 mRNA 3′-UTR and inhibits its translation[5]. We further found that NDR1 directly binds to the intergenic region of miR146a and inhibits NF-κB-mediated miR146a expression, thereby leading to enhanced STAT1 expression. Even though the antiviral effect of NDR1 is abolished in miR146a- or NDR1 DBS-deficient cells, we cannot exclude the possibility that other protein(s) or noncoding RNAs molecule(s) could be important for NDR1 as its functional cooperator. The integrative genomics, proteomics and epigenomics analysis approach could be further used to explore this possibility.

Intriguingly, we found that STAT1 also binds to the miR146a promoter and inhibits miR146a expression. IRAK1, IRAK2, TRAF6, and STAT1 have been identified as targets of miR146a[5,7,8,28]. We found that STAT1 was more sensitive to miR146a-mediated inhibition than IRAK1 and TRAF6, which may be an underlying mechanism by which NDR1 specifically upregulates STAT1 but does not affect the other targets at basal level. Moreover, the mutual inhibition between miR146a and STAT1 may be the underlying mechanism by which STAT1, but not IRAK1 or TRAF6, was downregulated by the low dose of miR146a. However, additional underlying mechanisms may exist that NDR1 downregulates STAT1 expression via miR146a, but does not affect IRAK1 and TRAF6 expression. Alternatively, there may be mechanisms that do not affect STAT1 expression but antagonize the downregulation of IRAK1 and TRAF6 by miR146a.

According to publicly available gene array data, NDR1 mRNA is downregulated in RAW264.7 cells infected with *Toxoplasma*

*gondii* (GEO accession GSE55298) or murine norovirus (GEO accession GSE12518)[35,36]. Here, we observed that NDR1 mRNA levels were significantly decreased in peripheral blood samples from patients infected with RSV compared with those of healthy controls. Furthermore, the mRNA and protein levels of NDR1 were decreased in macrophages infected with RNA viruses (RSV and VSV), DNA viruses (HSV and MHV68), and LM. These data reveal a mechanism whereby virus may escape immune surveillance by downregulating NDR1 expression through type I IFN feedback.

It has been reported that NDR1 kinase promotes $G_1/S$ cell cycle transition via directly phosphorylating p21 and subsequently reducing p21 stability[10]. Joffre et al.[37] have reported that NDR1 positively regulating autophagy via supporting the interaction of the exocyst component Exo84 with Beclin1 and RalB in a kinase dependent manner. We recently identified that NDR1 acts as an adapter to regulate IL17-induced inflammation in HeLa cervical carcinoma cells and mouse embryonic fibroblasts (MEFs)[38]. Here, we demonstrated that NDR1 is required for antiviral innate immune response via acting as a transcriptional regulator to bind to miR146a intergenic region, dampening miR146a transcription and subsequently enhancing STAT1 translation, which is in a kinase-independent manner. In our previous study, we found NDR1 interacts with TRAF3 and prevent its binding to IL17 receptor (IL-17R), which promotes the formation of an IL-17R-Act1-TRAF6 complex and downstream signaling[38]. We have examined the association of NDR1 with TRAF3 in macrophages since TRAF3 plays an important role in the antiviral immune response and found that NDR1 does not interact with TRAF3 in macrophages (data not shown). On the other hand, the level of STAT1 expression was comparable in NDR1-deficient and WT MEFs cells (data not shown). Therefore, NDR1, a generalist, plays roles in diverse cells and signaling pathways via different mechanisms.

Together, our findings revealed a novel regulator of innate immunity, NDR1, which is downregulated by infection with various viruses and regulates IFN signaling by maintaining STAT1 translation. Given that NDR1 is required for STAT1 expression and HCV clearance in liver cells, NDR1 may be a potential target for the clinical treatment of HCV infection.

## Methods

**Mice.** $NDR1^{-/-}$ mice a C57BL/6J background were gifts from Professor Brian Hemmings at the Friedrich Miescher Institute for Biomedical Research. Eight-week-old male groups of littermate mice were used in the in vivo experiments. Genotyping of WT and knockout mice was performed by PCR analysis of DNA isolated from the tail using the following three primers: 5′-GTACATTAGGTAA GACTTGAGG-3′, 5′-CTAGCTCATCCAGCCATGTG-3, and 5′-GCAGCGCAT CGCCTTCTATC-3′. All mice were maintained under specific-pathogen-free (SPF) conditions in the University Laboratory Animal Center. All animal experimental procedures were approved by the Animal Review Committee at Zhejiang University School of Medicine and were in compliance with institutional guidelines.

**Cells and reagents.** HEK293, L929, and RAW264.7 were obtained from American Type Culture Collection. Huh7 cells were gift from Dr. Xiaoben Pan (Peking University Hepatology Institute). *IFNαR* and *STAT1*-deficient immortal BMDMs were gift from Dr. Genhong Cheng (University of California, Los Angeles). Mouse primary PMs were prepared from female C57BL/6J mice (5–6-week old) through intraperitoneal injection with thioglycolate. The cells were cultured at 37 °C under 5% $CO_2$ in RPMI-1640 medium (Invitrogen) supplemented with 10% heat-inactivated FBS (Biology Industries), 100 U ml$^{-1}$ penicillin (Beyotime) and 100 μg ml$^{-1}$ streptomycin (Beyotime). Primary bone marrow cells were collected from WT and $NDR1^{-/-}$ mice to prepare bonemarrow-derived macrophages, which were cultured at 37 °C under 5% $CO_2$ in RPMI-1640 medium (Invitrogen) supplemented with 10% heat-inactivated FBS (Biology Industries) FBS, 20 ng ml$^{-1}$ M-CSF (PeproTech), 100 U ml$^{-1}$ penicillin (Beyotime) and 100 μg ml$^{-1}$ streptomycin (Beyotime). Mouse recombinant IFNβ, IFNγ and human recombinant IFNα were purchased from PeproTech. Poly (I:C), poly (dA:dT) and anti-flag mouse magnetic (M2) beads were from Sigma-Aldrich. Information about the antibodies used in this study is provided in Supplementary Table 1.

**ChIP-qPCR assay**. ChIP assays were performed with an EZ-Magna ChIP A/G kit (Millipore) according to the manufacturer's instructions. At least $1 \times 10^6$ cells were fixed with 1% formaldehyde for cross-linking of DNA with proteins, after which the cells were lysed with lysis buffer and nuclear lysis buffer. After sonication, DNA was immunoprecipitated overnight at 4 °C with 2 μg of an anti-NF-κB antibody, anti-STAT1 antibody, anti-H3K4me3 antibody, anti-H3K27me3 antibody or negative control immunoglobulin, and protein A/G beads or antiflag (M2) beads (Sigma-Aldrich). The beads, antibody, chromatin mixture was washed, and elution from the beads was performed with the supplied buffers. Then, cross-linking was reversed, and the samples were analyzed via quantitative PCR. The oligonucleotide sequences used for quantitative PCR are listed in Supplementary Table 2. ChIP-seq experiments were performed by GENEWIZ lnc.

**DNA pull-down assay**. DNA probes labeled with biotin were synthesized by BioSune Biotech. The sequence of the DNA region in the miR146a intergenic region that was recognized and bound by NDR1 contained biotin: biotin-miR146a intergenic A–F (biotin-AAGAATCCTGAGATACAATCAAAAAAACAA), miR146a intergenic A–R (TTGTTTTTTTGATTGTATCTCAGGATTCTT), biotin-miR146a intergenic B–F (biotin-TTGGCGGGCCGCAGACAACCGGCCACCATC), miR146a intergenic B–F–R (GATGGTGGCCGGTTGTCTGCGGCCCGCCAA), biotin-DNA negative control-F (biotin-AGGGCTCCTCATGCCTGGGATTGGGGTTTC), and biotin-DNA negative control-R (GAAACCCCAATCCCAGGCATGAGGAGCCCT). Single strands of DNA were thermally annealed to form dsDNA prior to pull-down experiments. Myc-NDR1-his was transiently over-expressed in HEK293 cells and purified using a Ni-NTA-Sefinose column (Sangon Biotech) according to the manufacturer's protocol. Then, the myc-NDR1-his protein in DNA pull-down buffer (10 mM HEPES, 150 mM NaCl, 1 mM MgCl$_2$, 0.5 mM EDTA, 0.5 mM DTT, 0.1% (vol/vol) NP40, and 10% (vol/vol) glycerol) or a cell lysate was mixed with the biotinylated DNA probe and incubated for 30 min at 25 °C. Streptavidin Sepharose beads (Thermo) were then added, and samples were incubated for another 30 min. The beads were washed three times with DNA pull-down buffer and resolved via SDS-PAGE with 2× SDS loading buffer, then subjected to immunoblotting.

**Human peripheral blood samples**. A total of 77 peripheral blood samples were collected from children with bronchiolitis hospitalized in Children's Hospital, Zhejiang University School of Medicine, and 40 healthy children as a control group. Nasopharyngeal aspirates were subjected to RSV antigen tests to confirm the RSV infection and metapneumovirus, parainfluenza virus, influenza virus, and adenovirus antigens were tested at the same time to exclude other common respiratory tract virus infections. Meanwhile, other microbiological tests were performed to exclude other respiratory tract infections and tuberculosis, including blood cultures, protein-purified derivative test, and serology for *Legionella pneumophila*, *Mycoplasma pneumonia*, and *Chlamydia pneumoniae*. No other pathogens were found by these tests. Peripheral blood samples were obtained from the patients on admission. This study was approved by the ethics committee of the Children's Hospital, Zhejiang University School of Medicine, and was in compliance with institutional guidelines. Written informed consent was obtained from at least one guardian of each patient before enrollment. The detailed patient's information is listed in Supplementary Table 3.

**Statistical analysis**. Statistical analysis was performed with Prism v5.0 (Graphpad Software). For in vivo experiments, values are expressed as the mean ± SD of $n$ animals and data shown are representative of three independent experiments. Mouse survival curves and statistics were analyzed with the Mantel–Cox test. The level of statistically significant difference was defined as $p < 0.05$. No data points or mice were excluded from the study. No randomization or blinding was used.

**Source data**. The uncropped scans of the western blottings were provided as the Supplementary Figs. 9–13.

**Ethical approval**. All animal experimental procedures were approved by the Animal Review Committee at Zhejiang University School of Medicine and were in compliance with institutional guidelines (No. ZJU20170074). The human work was approved by the ethics committee of the Children's Hospital, Zhejiang University School of Medicine, and was in compliance with institutional guidelines (No. 2017-016). Written informed consent was obtained from at least one guardian of each patient before enrollment.

**Data availability**. All relevant data are available from the authors upon request. The ChIP-seq datasets were deposited in gene expression omnibus (GEO), with an accession number of SRP149759.

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

## Acknowledgments

We thank Dr. Jinghao Sheng and Dr. Rongpan Bai (Zhejiang University) for excellent technical assistance and Dr. Huazhang An for critical reading of the manuscript. This work was supported by grants from the National Basic Research Program of China (973) (2014CB542101), the National Natural Science Foundation of China (31770932, 31570864, 81430040, and 81571738).

## Author contributions

X.W designed research; Z.L., Q.Q., C.W., H.L., J.Sh., M.G, Y.Y., C.M., W.L., Y.Zo. and Y.Zh. performed research; F.M. and J.Su. contributed reagent; X.W. and Z.L. analyzed data and wrote the paper.

## Additional information

**Competing interests:** The authors declare no competing interests.

