## [Peer Review File · Nature Communications]

Reviewers' comments:

Reviewer #1 (IFN signalling, miRNA)(Remarks to the Author):

The authors ascribed a new role for NDR1 in regulating innate immune response that is independent of its kinase activity, and explored the underlying molecular mechanism. The evidences came from both in vitro and in vivo studies with a combination of various technical approaches. The data for NDR1 function looked convincing with consistent conclusion from both gain-of-function and loss-of-function experiments, however there are some major concerns on the molecular mechanism part:

1.The authors identified a NDR1 binding site in mouse miR-146a intergenic region. However it's ~70kb away from the miR-146a locus. There is no clear evidence on how the sort of distal regulation might occur.

2.The authors examined miR-146a level and found NDR1 inhibits the miRNA expression. However they used an about 50-nt long oligo as the specific primer for the miRNA reverse transcription, which I don't see how it can detect the mature miRNA. The sequence also isn't complementary to miR-146a precursor, so all the miR-146a expression data in the manuscript is questioned.

3.I am unaware of any published characterization of mouse miR-146a promoter, the authors also didn't cite any papers about it. Yet they examined the binding sites of NF-kB and STAT1 ~10 kb upstream of miR-146a and claimed they are in miR-146a promoter. Where is the transcription start site of pri-miR-146a? On the other hand, since the function of NDR1 is conserved, and the human miR-146a regulatory region is well characterized, it will be more clear if similar findings can be made in a human system.

4. While there might be other mechanisms to ensure specific regulation of STAT1 by miR-146a, with no effect on other canonical targets such as IRAK1 and TRAF6, the latter being more sensitive to changes in miR-146a levels. The Western blot shown in Fig. 6e for is of poor quality to support the idea. Did the authors see changes in IRAK1 and TRAF6 expression with miR-146a mimic or inhibitor? if so those should be run in parallel as control.

5.The authors observed on effect of NDR1 on IRF3 and NF-kB activity, however NDR1 did substantially regulate IRF3 and NF-kB-dependent interferon or inflammatory cytokine expression, so what's the interpretation of such data? what is then the mechanism for the regulation?

other comments:

1.Because the authors stated that their patients only have RSV infection, the test reports that the authors used to exclude the possibility of other viral and bacterial infection of the patients should be included in the supplementary data to show the audience the method they use and the results they got. And it is better to provide the exact patients information, such as their age, their gender, and when the blood samples were collected (early stage or late stage of the infection), the treatment, how long had the patients been treated when they had the blood draw. In the following sentence, they authors made a mistake by saying that PBMCs are some kind of macrophages.['NDR1 expression was decreased in various macrophages infected with VSV, including human primary peripheral blood mononuclear cells (PBMCs)']

2.The statement in the following sentence that by poly(I:C) and poly(dA:dT) they rule out the possibility that virulence factors drive the suppression of NDR1 is not appropriate here.['In addition, transfection of poly(I:C) and poly(dA:dT) nucleic acid mimics in PMs also downregulated NDR1 expression (Supplementary Fig. 1b), thus excluding the possibility that virulence factors might drive the suppression of NDR1 expression after viral infection.']

3. It looks like overexpression of NDR1 did increase the p-p65 and p-IRF3, and in Fig4b, based on

that the bands for p-p65 and p-IRF3 and the total protein level of p65 and IRF3 seems lower in NDR1 overexpression groups.

Additionally, the following paper shows that miR-146a negative regulates RIG-I signaling pathway and downstream production of IFN β , which is known that IFN β expression is mediated by IRF3, so how to explain that NDR1 inhibits miR-146a, but have no effects on IRF3 mediated luciferase expression, and why there was only 'slightly' effects on NF κ B activation.

By IFNAR $^{-/-}$ or STAT1 $^{-/-}$ KO with knockdown of NDR1 can't prove that NDR1 functioned only downstream of IFN in RIG-I mediated anti-viral process, because NDR1 affects the production of IFN as shown in the results and that indicates NDR1 could function by regulating RIG-I signaling pathway, and this is upstream of IFN, and the authors can't exclude this possibility by IFNAR $^{-/-}$ or STAT1 $^{-/-}$ KO cells.

[J Immunol. 2009 Aug 1;183(3):2150-8. doi: 10.4049/jimmunol.0900707. Epub 2009 Jul 13. MicroRNA-146a feedback inhibits RIG-I-dependent Type I IFN production in macrophages by targeting TRAF6, IRAK1, and IRAK2.]

4. The following sentence is opposite to what have been presented in previous figure, that NDR1 $^{-/-}$ mice did have less TNF α and IL6.

[NDR1 did not affect the enrichment of NF- κ B at the promoters of these genes, thus possibly explaining why we observed no effects of NDR1 on these downstream mediators in earlier experiments.]

5. About why NDR1 affect miR-146a and STAT1 expression, but did not affect other targets of miR-146a, the authors explanation in the discussion is confusing.

6. In the abstract, the authors states that downregulation of NDR1 might be a escape mechanism of virus is not appropriate.

7. The targeting of STAT1 by miR-146a was initially reported in Tang et al, Arth Rheum, 2009, which should be cited.

8. The authors indicated "mock" for the control of overexpression experiments, it's not clear it's just parental cells, or empty vector transfection, or stable line expressing a non-relevant protein (e.g., GFP)? Nevertheless, the last one might be better, to exclude that expression of any exogenous DNA changed the cellular response.

Reviewer #2 (miR146a, cross-regulation)(Remarks to the Author):

The manuscript by Liu et al investigates a role for the kinase NDR1 during viral infection. The authors demonstrate that during viral or bacterial infection or type I IFN signaling NDR1 expression is repressed. To assess function, NDR1 $^{-/-}$ mice are used and shown to have increased susceptibility to viral infection. Mechanistically the study provides evidence that NDR1 directly binds to the miR-146a gene and represses its expression by hindering NF- κ B transactivation. This, in turn, leads to reduced miR-146a repression of Stat1 which is expressed at higher levels and drives the IFN response.

Overall, the work is well done and the results are generally convincing. Conceptually, this study does offer novel insight into how an NDR1 regulates the antiviral response by regulating a miR-146a-Stat1-ISG axis. However, the following points should be addressed before the work is ready for publication.

1. How is NDR1 downregulated during viral infection? The paper starts off with this observation but does not explore the mechanistic basis for this. This should be addressed with new experiments.
2. Materials and Methods are lacking and should be expanded and greater detail provided. For

instance, how were miR-146a $-/-$ 293T and L929 cells developed/obtained?

3. How does the proposed mechanism of NDR1 function relate to its roles in other contexts reported by the same authors and others? The discussion should be expanded to cover these considerations.
4. The authors show relatively weak evidence for decreased levels of STAT1 in virally infected NDR1 $-/-$ mice. Further, they do not show evidence of enhanced miR-146a levels in these animals. Additional data from in vivo infection models showing enhanced miR-146a and decreased STAT1 in NDR1 $-/-$ mice would further solidify the story.
5. The authors should show that the siNDR1 treatment they use for multiple experiments also leads to increased expression of miR-146a, present this in the main figures.
6. Could the proposed NDR1 binding site in the intergenic region of miR-146a be targeted by CRISPR to provide additional evidence for that being the critical binding site?
7. For figure 3e, please label to clarify that the survival curve is for HSV infection
8. For figure 4b, there are two names above the set of blots on the far right. Also, figure 4b and figure 5 use different names for kinase-inactivated NDR1 mutants. Please clarify the names of these mutants in the figures.
9. For figure 6g, please show a schematic showing what sections of the intergenic region are amplified by which primers in the figure. This will help in evaluating the data.
10. For figures 7c, 7d, and 8c, please label the figures to clarify that the data is from NFkB pull-downs.
11. I would suggest that the authors move the Listeria in vivo data from figure 5 to figure 3 or move it to supplemental.

Reviewer #3 (Innate signalling, RNA sensing)(Remarks to the Author):

In the manuscript entitled, "Downregulation of NDR1 due to viral infection inhibits the innate immune response by initiating an miR146a-STAT1 feedback loop" Liu et al. describe their findings concerning a molecular circuit involving NDR1, STAT1, and miR146a. The study begins by demonstrating a modest increase in NDR1 expression in response to a diverse family of viruses or treatment of interferon. The authors go on to demonstrate that this up regulation in NDR1 requires IFN signaling. Disruption of NDR1 signaling is found to promote the cellular antiviral immune response to both RNA and DNA viruses both in vitro and in vivo. This activity appears to be the result of a specific nuclear activity of NDR1 which results in increased STAT1 expression. STAT1 elevation in response to NDR1 is kinase-independent and appears to be due to increased translation (as opposed to transcription or stability). The authors perform ChIP Seq on NDR1 and find that it engages a locus that encodes for miR146 (amongst other things). As this microRNA has already been demonstrated to target STAT1, the authors investigate this further and find that miR-146 is two fold higher in the absence of NDR1 and go on to show that knockdown of NDR1 in miR146 knockout cells has little impact on STAT1-dependent ISGs. The authors then conclude their paper but looking at the dynamics of NDR1/IFN/miR146 and virus infection using an HCV model.

Overall, this review finds much of the data concerning NDR1 and STAT1 compelling. However, the data implicating miR146 in this biologic circuit is less convincing. As miR146 has already been demonstrated to suppress STAT1, one would expect much of the data presented regardless of the role of NDR1. Moreover, there are some aspects to the proposed mechanism which make it difficult to comprehend how this model could be correct. My two major concerns are listed below.

1. Timing. In figure 7g the authors demonstrate the impact of silencing NDR1 on ISGs within 8hrs of infection. Given that the half life of miRNAs are 36-48hrs, how does engagement of the gene have an impact so quickly? Can the authors generate a miR-146 transgenic cell and determine

whether the NDR1 phenotype is lost? This should not be done with mimetics as this will undoubtedly result in ultra physiological levels of miR146.

2. Copy number. If the authors are proposing an involvement in miR146 in this IFN circuit, they must determine copy number per cell and demonstrate a loss of miRNA processing at the level of northern blot should this biology really be responsible for the changes in STAT1 expression. A 2 fold increase does not seem adequate to explain the phenotype observed.

We thank the reviewers for their comments and suggestions, and have revised the manuscript and added additional experimental data in response.

According to the suggestions and comments, we tried our best to address the questions point-to-point raised by the reviewers and made revisions of the manuscript, and the changes are marked in red in the revised manuscript. We believe that we have addressed all of the problems brought up in the review.

To Reviewer 1:

Thank you very much for your comments and suggestions regarding our manuscript entitled “Downregulated NDR1 protein kinase inhibits innate immune response by initiating an miR146a-STAT1 feedback loop” (NCOMMS-17-12910). According to your suggestions, we have added some new data and re-organized the manuscript. If you have any further questions, please inform us and we can discuss them further.

The authors ascribed a new role for NDR1 in regulating innate immune response that is independent of its kinase activity, and explored the underlying molecular mechanism. The evidences came from both in vitro and in vivo studies with a combination of various technical approaches. The data for NDR1 function looked convincing with consistent conclusion from both gain-of-function and loss-of-function experiments, however there are some major concerns on the molecular mechanism part:

1. Question: The authors identified a NDR1 binding site in mouse miR-146a intergenic region. However it's ~70kb away from the miR-146a locus. There is no clear evidence on how the sort of distal regulation might occur.

Response: It has been reported that some enhancers can interact with the promoters which are more than 1Mb away and regulate gene transcription^{1,2}. NDR1 might bind to miR146a intergenic region to regulate miR146a expression though miR146a intergenic region is 70kb away from the miR146a locus.

2. Question: The authors examined miR-146a level and found NDR1 inhibits the

miRNA expression. However they used an about 50-nt long oligo as the specific primer for the miRNA reverse transcription, which I don't see how it can detect the mature miRNA. The sequence also isn't complementary to miR-146a precursor, so all the miR-146a expression data in the manuscript is questioned.

Response: The “stem loop” specific primer (5'-GTCGTATCCAGTGCAGGGTCC GAGGTATTCGCACTGGATACGACAACCCA-3') was used for mature miR146a reverse transcription in our study. The 6 bases marked in red are reverse complementary to the bases marked in red at 3' terminal of has- and mmu-miR-146a-5p (5'-UGAGAACUGAAUCCAUGGGUU-3'). In addition, Hou et al used a similar stem loop primer (5'-GTCGTATCCAGTGCAGGGTCCGAGGTA TTCGCACTGGATACGACAACCC-3') for mature miR146a reverse transcription³.

3. Question: I am unaware of any published characterization of mouse miR-146a promoter, the authors also didn't cite any papers about it. Yet they examined the binding sites of NF-κB and STAT1 ~10 kb upstream of miR-146a and claimed they are in miR-146a promoter. Where is the transcription start site of pri-miR-146a? On the other hand, since the function of NDRI is conserved, and the human miR-146a regulatory region is well characterized, it will be clearer if similar findings can be made in a human system.

Response: Yes, there is no published characterization of mouse miR146a promoter. Jayeeta et al have reported that NF-κB binds to regulatory sequences (-10680~-11120) in the upstream of miR146a in mouse striatal cells and induces the transcription of miR146a⁴. We have changed “mouse miR146a promoter” into “mouse miR146a regulatory sequences” in the current manuscript.

Taganov et al have reported the transcription start site of pri-miR-146a in human cells⁵. The schematic diagram of human miR-146a genomic loci is shown in **Response Figure 1**, in which the transcription start site of pri-miR-146a is marked in red.

Response Figure 1. The schematic diagram of miR-146a genomic loci on human chromosomes 5.

According to your suggestion, we have knocked down NDR1 expression in human acute monocytic leukemia cell line Thp1 cells. As shown in Supplementary Fig. 4f in the revised manuscript, silencing of NDR1 resulted in reduced STAT1 expression, enhanced miR146a expression (Fig. 7f) and less NF-κB enrichment on miR146a promoter (Supplementary Fig. 7g). Consistently, overexpression of NDR1 in Thp1 cells lead to enhanced STAT1 expression (Supplementary Fig. 4g), less NF-κB enrichment on miR146a promoter (Fig. 7f) and decreased miR146a expression (Supplementary Fig. 7b). We also demonstrated that silencing of STAT1 promoted NF-κB to bind to miR146a promoter (Fig. 8d) and increased miR146a expression (Fig. 8b) in human Thp1 cells in the current manuscript. Taken together, these results indicate that the inhibitory effect of NDR1 on NF-κB-mediated miR146a transcription is conserved in mouse and human.

4. Question: *While there might be other mechanisms to ensure specific regulation of STAT1 by miR-146a, with no effect on other canonical targets such as IRAK1 and TRAF6, the latter being more sensitive to changes in miR-146a levels. The Western blot shown in Fig. 6e for is of poor quality to support the idea. Did the authors see changes in IRAK1 and TRAF6 expression with miR-146a mimic or inhibitor? if so those should be run in parallel as control.*

Response: Yes, you are right. We could not exclude other mechanisms underlying the specific regulation of STAT1 by miR146a. We had mentioned this point on the page

24 which was marked in red in the current manuscript.

We have repeated the experiment for Fig.6e and the results were shown in Fig. 6e in the current manuscript. The effect of miR-146a mimic or inhibitor on the expression IRAK1 and TRAF6 was shown in Supplementary Fig. 7n,o in the current manuscript.

5. Question: The authors observed no effect of NDR1 on IRF3 and NF- κ B activity, however NDR1 did substantially regulate IRF3 and NF- κ B-dependent interferon or inflammatory cytokine expression, so what's the interpretation of such data? what is then the mechanism for the regulation?

Response: It has been reported that after sensing viral nucleic acids, PRRs, such as RIG-I, activates IRF3 and NF- κ B, which cooperatively induce type I interferon (IFN) expression. Type I interferon then triggers the JAK-STAT signaling pathway and leads to amplification of ISGs (such as RIG-I) expression, which further make a positive feedback loop to RIG-I pathway⁶. NDR1 promoted RIG-I-N-induced IFN-sensitive response element (ISRE) reporter activation (Supplementary Fig. 4b) but not IRF3 reporter activation (Supplementary Fig. 4c), which indicated NDR1 promotes the activation of the type I IFN pathway but not the IRF3 pathway. More importantly, NDR1 significantly affected STAT1 expression with or without VSV infection (Fig. 4a,b and Supplementary Fig. 4a) and the effect of NDR1 on anti-virus immune response was abolished in STAT1 deficient macrophage (Fig. 5c,d). Actually, NDR1 slightly affected virus infection-induced IRF3 and NF- κ B activation, as well as the RIG-I expression (Fig. 4a,b and Supplementary Fig. 4a) at the 8 hours after VSV infection. Together, these results suggest that NDR1 has no effect on VSV induced NF- κ B and IRF3 activation, but enhances STAT1 expression and promotes the ISGs

induction including RIG-I, which further facilitates the activation of NF- κ B and IRF3 (Fig. 4b) and the production of type I interferon and inflammatory cytokines (Supplementary Fig. 2g) at the 8 hours after VSV infection. We have added the interpretation of such data on page 11 in the current manuscript.

Other comments

1. Question: Because the authors stated that their patients only have RSV infection, the test reports that the authors used to exclude the possibility of other viral and bacterial infection of the patients should be included in the supplementary data to show the audience the method they use and the results they got. And it is better to provide the exact patients information, such as their age, their gender, and when the blood samples were collected (early stage or late stage of the infection), the treatment, how long had the patients been treated when they had the blood draw.

Response: Thanks for your suggestion. The detailed patient's information has been shown in **Supplementary table**.

2. Question: In the following sentence, they authors made a mistake by saying that PBMCs are some kind of macrophages.['NDR1 expression was decreased in various macrophages infected with VSV, including human primary peripheral blood mononuclear cells (PBMCs) ']

Response: Sorry for the mistake. We have replaced the sentence with "NDR1 expression was decreased in human primary peripheral blood mononuclear cells (PBMCs) and various macrophages infected with VSV, including human acute monocytic leukemia cell line Thp1, murine bone marrow-derived macrophages (BMDMs) and RAW264.7" in the current manuscript.

3. Question: The statement in the following sentence that by poly(I:C) and poly(dA:dT) they rule out the possibility that virulence factors drive the suppression of NDR1 is not appropriate here.['In addition, transfection of poly(I:C) and poly(dA:dT) nucleic

acid mimics in PMs also downregulated NDR1 expression (Supplementary Fig. 1b), thus excluding the possibility that virulence factors might drive the suppression of NDR1 expression after viral infection.’]

Response: Yes, you are right. We have deleted this sentence “thus excluding the possibility that virulence factors might drive the suppression of NDR1 expression after viral infection.” in the current manuscript.

4. Question: *It looks like overexpression of NDR1 did increase the p-p65 and p-IRF3, and in Fig4b, based on that the bands for p-p65 and p-IRF3 and the total protein level of p65 and IRF3 seems lower in NDR1 overexpression groups.*

Response: Yes, you are right. We have changed the sentence “while exerting no effect on the activation of the NF- κ B, MAPK, IRF pathways” with “while slightly promoted the activation of the NF- κ B and IRF3 pathways and the expression of RIG-I at the 8 hours after VSV infection” on page 11 in the current manuscript.

5. Question: *The following paper shows that miR-146a negative regulates RIG-I signaling pathway and downstream production of IFN β , which is known that IFN β expression is mediated by IRF3, so how to explain that NDR1 inhibits miR-146a, but have no effects on IRF3 mediated luciferase expression, and why there was only ‘slightly’ effects on NF- κ B activation. [J Immunol. 2009 Aug 1;183(3):2150-8. doi: 10.4049/jimmunol.0900707. Epub 2009 Jul 13. MicroRNA-146a feedback inhibits RIG-I-dependent Type I IFN production in macrophages by targeting TRAF6, IRAK1, and IRAK2.]*

Response: Our results suggested that NDR enhances STAT1 expression and promotes the ISGs induction including RIG-I, which further facilitates the activation of NF- κ B, IRF3 (**Fig. 4b**) and the production of type I interferon and inflammatory cytokines (**Supplementary Fig. 2g**) at the 8 hours after VSV infection. NDR1 did not affect RIG-I-N-induced IRF3 reporter activation (**Supplementary Fig. 4c**) but IFN-sensitive response element (ISRE) reporter activation (**Supplementary Fig. 4b**), which indicates that NDR1 promotes the activation of the type I IFN pathway but not

RIG-I pathway. Hou et al had reported miR146a inhibits VSV-induced Type I IFN production by targeting TRAF6, IRAK1 and IRAK2. However, they showed miR146a inhibits VSV induced IFN expression at 18 hour or later after VSV infection (**Response Figure. 2**)³, which might partially result from the IFN feedback loop. Moreover, we found that NDR1 had no effect on the expression of IRAK1 and TRAF6 (**Fig. 6e**). The mutual inhibition between miR146a and STAT1 might be the underlying mechanism by which NDR1 specifically upregulates STAT1 but does not affect the expression of IRAK1 and TRAF6 in the resting state.

Response Figure 2. Mouse peritoneal macrophages were transfected with control mimics or miR-146a mimics (B), control inhibitor or miR-146a inhibitor (C) as indicated. After 48 h, cells were infected by VSV at MOI 10 for indicated time. IFN-mRNA expression (left) was measured by q-PCR and normalized to the expression of beta-actin in each sample. IFN-beta in supernatants (right) was measured by ELISA.

6. Question: *By IFNAR^{-/-} or STAT1^{-/-} KO with knockdown of NDR1 can't prove that NDR1 functioned only downstream of IFN in RIG-I mediated anti-viral process, because NDR1 affects the production of IFN as shown in the results and that indicates NDR1 could function by regulating RIG-I signaling pathway, and this is upstream of IFN, and the authors can't exclude this possibility by IFNAR^{-/-} or STAT1^{-/-} KO cells.*

Response: You are right. It can't prove that NDR1 functioned only downstream of IFN in RIG-I mediated anti-viral process via IFNAR^{-/-} or STAT1^{-/-} KO with knockdown of NDR1. However, NDR1 did not affect RIG-I-N-induced IRF3 reporter activation (**Supplementary Fig. 4c**) but IFN-sensitive response element (ISRE)

reporter activation (Supplementary Fig. 4b), which indicates that NDR1 promotes the activation of the type I IFN pathway but not RIG-I pathway.

7. Question: The following sentence is opposite to what have been presented in previous figures, that NDR1^{-/-} mice did have less TNF α and IL6. [‘NDR1 did not affect the enrichment of NF- κ B at the promoters of these genes, thus possibly explaining why we observed no effects of NDR1 on these downstream mediators in earlier experiments.’]

Response: Thanks for your suggestion. The sentence “NDR1 did not affect the enrichment of NF- κ B at the promoters of these genes, thus possibly explaining why we observed no effects of NDR1 on these downstream mediators in earlier experiments” was not accurate. We have changed it into “NDR1 did not affect the enrichment of NF- κ B at the promoters of these genes, thus possibly explaining why we observed no effect of NDR1 on these downstream mediators at 4 hour after VSV infection (Fig. 2b and Supplementary Fig. 2d)” in the current manuscript.

ChIP assay showed NDR1 did not affect the enrichment of NF- κ B at the promoters of these genes (Supplementary Fig. 7j.k). NDR1 knockdown or knockout had no effect on IL6 and TNF α production at 4 hour after VSV infection but decreased IL6 and TNF α production at 8 hour upon VSV infection (Fig. 2b and supplementary Fig 2d). Moreover, NDR1 slightly affected virus infection-induced NF- κ B activation, as well as the RIG-I expression (Fig. 4a,b and Supplementary Fig. 4a) at the 8 hours upon VSV infection. Together, these results suggest that NDR1 affected the VSV-triggered ISGs expression such as RIG-I, which further regulates NF- κ B activation and the production of inflammatory cytokines. Consistently, the NDR1^{-/-} mice had less TNF α and IL6 at 12 hour after VSV infection (Fig. 3a and Supplementary Fig. 3a-d).

5. Question: About why NDR1 affect miR-146a and STAT1 expression, but did not affect other targets of miR-146a, the author's explanation in the discussion is confusing.

Response: In this study, we demonstrated that NDR1 directly binds to the intergenic region of miR146a and inhibits NF- κ B-mediated miR146a expression; thereby leading to enhanced STAT1 expression. However, NDR1 had no effect on the expression of IRAK1 and TRAF6, the other targets of miR146a. We further found that STAT1 also binds to the miR146a promoter and inhibits miR146a expression. NDR1 deficient macrophages expressed only about 2-fold more miR146a than WT macrophage (**Fig. 7c**), the mutual inhibition between miR146a and STAT1 may be the underlying mechanism by which NDR1 specifically upregulates STAT1 but does not affect the other targets. However, additional underlying mechanisms may exist that NDR1 downregulates STAT1 expression via miR146a but does not affect IRAK1 and TRAF6 expression.

6. Question: In the abstract, the authors states that downregulation of NDR1 might be a escape mechanism of virus is not appropriate.

Response: Thanks for your advice. We had corrected the sentence in the current manuscript as below “Together, our result identify NDR1 promoting STAT1 translation as an essential event for IFN-dependent antiviral immune response and indicates the important role of NDR1 in controlling viral infections.”

7. Question: The targeting of STAT1 by miR-146a was initially reported in Tang et al, Arth Rheum, 2009, which should be cited.

Response: Thanks for your suggestion. We have cited this paper in the current manuscript.

8. Question: The authors indicated "mock" for the control of overexpression experiments, it's not clear it's just parental cells, or empty vector transfection, or stable line expressing a non-relevant protein (e.g., GFP)? Nevertheless, the last one

might be better, to exclude that expression of any exogenous DNA changed the cellular response.

Response: "Mock" indicates empty vector transfection. We have changed "mock" into "flag-none" in the current manuscript.

To Reviewer 2:

Thank you very much for your comments and suggestions regarding our manuscript entitled “Downregulated NDR1 protein kinase inhibits innate immune response by initiating an miR146a-STAT1 feedback loop” (NCOMMS-17-12910). According to your suggestions, we have added some new data and re-organized the manuscript. If you have any further questions, please inform us and we can discuss them further.

The manuscript by Liu et al investigates a role for the kinase NDR1 during viral infection. The authors demonstrate that during viral or bacterial infection or type I IFN signaling NDR1 expression is repressed. To assess function, NDR1-/- mice are used and shown to have increased susceptibility to viral infection. Mechanistically the study provides evidence that NDR1 directly binds to the miR-146a gene and represses its expression by hindering NF- κ B transactivation. This, in turn, leads to reduced miR-146a repression of Stat1 which is expressed at higher levels and drives the IFN response.

Overall, the work is well done and the results are generally convincing. Conceptually, this study does offer novel insight into how NDR1 regulates the antiviral response by regulating a miR-146a-Stat1-ISG axis. However, the following points should be addressed before the work is ready for publication.

1. Question: How is NDR1 downregulated during viral infection? The paper starts off with this observation but does not explore the mechanistic basis for this. This should be addressed with new experiments.

Response: Histone modifications play important roles in the control of gene transcription. H3K4 trimethylation (H3K4me3) is a marker of transcriptional activation⁷, while high levels of H3K27 trimethylation (H3K27me3) in the coding region generally correlate with transcription repression⁸. By ChIP assay, we found that H3K27me3 was upregulated in the promoter of NDR1 gene in Thp1 cells after VSV infection, whereas H3K4me3 remained constant throughout the infection with VSV (**Fig. 1g**). VSV infection-induced downregulation of NDR1 expression was abolished in IFN α R- or STAT1-deficient macrophages (**Fig. 1e,f** and **Supplementary Fig. 1e,f**).

Together, these data demonstrated that viral infection downregulates NDR1 expression via a IFN signaling-dependent mechanism and may also involve epigenetic modification of histone H3.

2. Question: Materials and Methods are lacking and should be expanded and greater detail provided. For instance, how were miR-146a -/- 293T and L929 cells developed /obtained?

Response: Thanks for your suggestion. We have expanded the details of materials and methods which were marked in red in the supplemental materials section in the current manuscript.

3. Question: How does the proposed mechanism of NDR1 function relate to its roles in other contexts reported by the same authors and others? The discussion should be expanded to cover these considerations.

Response: According to your suggestion, we have added some mechanisms of NDR1 function related to its roles in IL17 signaling pathway on page 24 to page 25 in the current manuscript as follows:

It has been reported that NDR1 kinase promotes G₁/S Cell Cycle transition via directly phosphorylating p21 and subsequently reducing p21 stability⁹. Carine Joffre et al have reported that NDR1 positively regulating autophagy via supporting the interaction of the exocyst component Exo84 with Beclin1 and RalB in a kinase dependent manner¹⁰. We recently identified that NDR1 acts as an adaptor to regulate IL17-induced inflammation in HeLa cervical carcinoma cells and mouse embryonic fibroblasts (MEFs)¹¹. Here, we demonstrated that NDR1 is required for anti-viral innate immune response via acting as a transcriptional regulator to bind to miR146a intergenic region, dampening miR146a transcription and subsequently enhancing STAT1 translation, which is in a kinase-independent manner. In our previous study, we found NDR1 interacts with TRAF3 and prevent its binding to IL17 receptor (IL-17R) which promotes the formation of an IL-17R-Act1-TRAF6 complex and downstream signaling¹¹. We have examined the association of NDR1 with TRAF3 in macrophages since TRAF3 plays an important role in the anti-viral immune response

and found that NDR1 does not interact with TRAF3 in macrophages (data not shown). On the other hand, the level of STAT1 expression was comparable in NDR1-deficient and WT MEFs cells (data not shown). Therefore, NDR1, a generalist, plays roles in diverse cells and signaling pathways via different mechanisms.

4. Question: The authors show relatively weak evidence for decreased levels of STAT1 in virally infected NDR1-/- mice. Further, they do not show evidence of enhanced miR-146a levels in these animals. Additional data from in vivo infection models showing enhanced miR-146a and decreased STAT1 in NDR1-/- mice would further solidify the story.

Response: According to your suggestion, we have tested the expression of STAT1 and mature miR146a in tissues obtained from wild-type and NDR1-deficient mice infected with VSV or HSV-1. NDR1 deficiency promoted the expression of mature miR146a (**Supplementary Fig. 7d,e**) and dampened STAT1 expression (**Supplementary Fig. 4h**) in various tissues including brain, lung, liver and spleen in both resting and infectious states.

5. Question: The authors should show that the siNDR1 treatment they use for multiple experiments also leads to increased expression of miR-146a, present this in the main figures.

Response: Thanks for your suggestion. We have knocked down NDR1 expression in mouse PMs, mouse immortalized BMDMs, L929, human PBMCs, Thp1 and HEK293 cells by its specific siRNA and tested the expression of miR146a expression. As shown in **Fig. 7e** in the current manuscript, silencing of NDR1 upregulated mature miR146a expression in these cells.

6. Question: Could the proposed NDR1 binding site in the intergenic region of miR-146a be targeted by CRISPR to provide additional evidence for that being the critical binding site?

Response: According to your suggestion, we designed seven gRNAs to target miR146a intergenic region (**Response Figure. 3**) and cloned them in the CRISPR-cas9 plasmids pEP-330x, which were transfected into L929 cells. As shown in **Response Figure. 4**. PCR analysis showed that none of the constructed CRISPR-cas9 plasmids generated double strand breaks (DSBs) in miR146a intergenic region.

```

AAAAAGAGGGCTCTCATGCCTGGGATTGGGGTTTCTGTTAGACAGTATTTGGCTCTTGGAAAGGAGCATTGTCAACCCAG
TGAGGAACATTTTTAAATGTTGAAATAAAAAATATGTCAAATAAAAAACACCAAACCATAAGCAATAGATTCAGCCAGAC
CGAAGGAAAAGTACCAGAGATAGAGGACATGGATGCGGAAATACTACACAAACACCGATAAATAAACTAAATAAAG
ATGACTGCAATGTCCAAGAATCCTGAGATACAATCAAAAAACAAAAGCAGAGTCCAAAAGACAAAGAACCTAAGAAT 1#
TTTTTAAAGGACAGACAGAGAGAATGTATTATTCATAAATTAAGAACAAAATAGACAGGTTAGAGGGTGCGCCAGAGA 2#
ACCGGACAGCTTCTGGGACGGGCGGAAGCACAGAGCCGCTGAGGCAGCACCCCTTGGCGGGCCGCAGACAACCGGCCACC 4#
ATCCGGACCAGAGGACAGGTTGCTGCCTGGCTTGGGAGGGCGGCTCAGCCTCAGCAGCAGCGGTTCGCCATCTTGGTTCCG 5# 6#
GGACTCAGCAGAACTGGGAAATTAGTCTGAACAGGTTAGAGGGTGCGCCAGAGAACCGGACAGCTTCTGGGACGGGCAG 7#
AAGCACAGAGCCGCTGAGGCAGCACCCCTTGGCGGGCCGAGACAGCCGGCCACCGTCTGGACCAGAGGACAGGTGCCCG
CCTGGCTTGGGAGGGCGGCTCAGCCTCAGCAGCAGCGGTTGCCATCTGGGTTCCAGGA

```

Response Figure 3. The sequence of murine miR146a intergenic region. The NDR1 DNA binding site is highlighted; the gRNAs targeted sequences were underlined, the primers for genotype identification are marked in green.

Response Figure 4. PCR analysis of genomic DNA from L929 cells transfected with CRISPR-cas9 plasmids containing gRNAs to target miR146a intergenic region.

7. Question: For figure 3e, please label to clarify that the survival curve is for HSV infection.

Response: Thanks for your suggestion; we have labeled the **Fig. 3e** in the current manuscript.

8. Question: For figure 4b, there are two names above the set of blots on the far right. Also, figure 4b and figure 5 use different names for kinase-inactivated NDR1 mutants. Please clarify the names of these mutants in the figures.

Response: Sorry for this mistake. We have replaced NDR1 mutant “S281A T444A” with “AA” in **Fig. 4b** in the current manuscript.

9. Question: For figure 6g, please show a schematic showing what sections of the intergenic region are amplified by which primers in the figure. This will help in evaluating the data.

Response: Thanks for your advice. We have added a schematic to show what sections of the intergenic region are amplified by which primers in **Fig. 6g** in the current manuscript.

10. Question: For figures 7c, 7d, and 8c, please label the figures to clarify that the data is from NF- κ B pulldowns.

Response: Thanks for your advice. We have added “IP: NF- κ B” on the top of figures in **Fig. 7f** and **Supplementary Fig. 7g-i** in the current manuscript.

11. Question: I would suggest that the authors move the Listeria in vivo data from figure 5 to figure 3 or move it to supplemental.

Response: According to your suggestion, we have moved Listeria *in vivo* data to supplemental (**Supplemental Fig. 5h,i**) in the current manuscript.

To Reviewer 3:

Thank you very much for your comments and suggestions regarding our manuscript entitled “Downregulated NDR1 protein kinase inhibits innate immune response by initiating an miR146a-STAT1 feedback loop” (NCOMMS-17-12910). According to your suggestions, we have added some new data and re-organized the manuscript. If you have any further questions, please inform us and we can discuss them further.

In the manuscript entitled, “Downregulation of NDR1 due to viral infection inhibits the innate immune response by initiating an miR146a-STAT1 feedback loop” Liu et al. describe their findings concerning a molecular circuit involving NDR1, STAT1, and miR146a. The study begins by demonstrating a modest increase in NDR1 expression in response to a diverse family of viruses or treatment of interferon. The authors go on to demonstrate that this up regulation in NDR1 requires IFN signaling. Disruption of NDR1 signaling is found to promote the cellular antiviral immune response to both RNA and DNA viruses both in vitro and in vivo. This activity appears to be the result of a specific nuclear activity of NDR1 which results in increased STAT1 expression. STAT1 elevation in response to NDR1 is kinase-independent and appears to be due to increased translation (as opposed to transcription or stability). The authors perform ChIP Seq on NDR1 and find that it engages a locus that encodes for miR146 (amongst other things). As this microRNA has already been demonstrated to target STAT1, the authors investigate this further and find that miR-146 is two fold higher in the absence of NDR1 and go on to show that knockdown of NDR1 in miR146 knockout cells has little impact on STAT1-dependent ISGs. The authors then conclude their paper but looking at the dynamics of NDR1/IFN/miR146 and virus infection using an HCV model.

Overall, this review finds much of the data concerning NDR1 and STAT1 compelling. However, the data implicating miR146 in this biologic circuit is less convincing. As miR146 has already been demonstrated to suppress STAT1, one would expect much of the data presented regardless of the role of NDR1. Moreover, there are some aspects

to the proposed mechanism which make it difficult to comprehend how this model could be correct. My two major concerns are listed below.

Response: In our study, we identified mutual inhibition between miR146a and STAT1, such that miR146a downregulates STAT1 translation by targeting the 3'-UTR of STAT1 mRNA, and STAT1 binds to the miR146a promoter and inhibits NF- κ B-mediated miR146a transcription. We further confirmed the role of miR146a in the anti-virus immune response in the pre-miR146a overexpressed and mir146R deficient cells. The delicate mechanism how miR146a is regulated need to be further investigated, such as the identification of the promoter/enhancer region and the transcription start site of mouse miR146a.

1. Question: Timing. In figure 7g the authors demonstrate the impact of silencing NDR1 on ISGs within 8hrs of infection. Given that the half life of miRNAs are 36-48hrs, how does engagement of the gene have an impact so quickly? Can the authors generate a miR-146 transgenic cell and determine whether the NDR1 phenotype is lost? This should not be done with mimetics as this will undoubtedly result in ultra physiological levels of miR146.

Response: Thanks for your advice. In **Fig. 7k** (figure 7g) in the current manuscript, WT and miR146a^{-/-} L929 cells were transfected with NDR1-specific or scrambled siRNA for 48 h and then infected with VSV. As shown in **Fig. 7e** and **Fig. 7j**, NDR1-specific siRNA transfection for 48h increased miR146a expression but inhibited STAT1 expression in L929 cells, which leads to the decreased ISGs induction triggered by following VSV infection.

According to your suggestion, we have generated stable pre-miR146a overexpressed L929 cells to determine whether the NDR1 phenotype is lost. As shown in **Fig. 7h** and **Supplementary Fig. 7p** in the current manuscript,

overexpression of NDR1 upregulated STAT1 expression and VSV-induced ISGs expression in control group but had no effect on STAT1 protein level and ISGs induction in pre-miR146a overexpressed L929 cells. Consistently, the inhibitory effect of NDR1 overexpression on VSV replication was abolished in the pre-miR146a overexpressed L929 cells (**Fig. 7i**).

2. Question: *Copy number. If the authors are proposing an involvement in miR146 in this IFN circuit, they must determine copy number per cell and demonstrate a loss of miRNA processing at the level of northern blot should this biology really be responsible for the changes in STAT1 expression. A 2 fold increase does not seem adequate to explain the phenotype observed.*

Response: Thanks for your suggestion, absolute quantitative real-time PCR assay of pri-miR146a was performed and the standard curve was generated by using pre-miR146a overexpressing plasmid, which consists of the stem loop structure and more than 200 base pairs of upstream and downstream flanking genomic sequence of miR146a. The primer of mouse pri-miR146a for PCR was designed according to the upstream and downstream flanking genomic sequence of mouse pre-miR146a which was obtained from DNA sequencing. As shown in **Fig. 7a,b**, the copy number per cell of pri-miR146a was higher in NDR1-deficient immortalized BMDMs (1.8×10^{-9} fmol/cell) than in WT immortalized BMDMs (0.5×10^{-9} fmol/cell) (**Fig. 7a**), while NDR1 overexpressed RAW264.7 cells showed lower pri-miR146a (1.6×10^{-10} fmol/cell) than the control transfectants (3.0×10^{-10} fmol/cell) (**Fig. 7b**).

Northern blot assay was performed to explore the effect of NDR1 on the miR146a processing. Considering that the radiative facility and probe for mouse miR146a are not available, we used the digoxigenin system to examine the miR146a processing in Thp1 cells. As shown in **Supplementary Fig. 7f**, only the expression of mature miR146a but not the expression of pri- and pre-miR146a was detectable in Thp1 cell and NDR1 knockdown significantly upregulated the expression of mature miR146a. Next, we investigated the effect of NDR1 on miR146a processing by testing pri-miR146a and pre-miR146a expression via QPCR. As expected, NDR1 knockout

in mouse PMs or knockdown in human PBMCs increased pri-, pre- and mature miR146a expression (**Fig. 7c and Supplementary Fig. 7a**). The overexpression of NDR1 in RAW264.7 or Thp1 inhibited pri-, pre- and mature miR146a expression (**Fig. 7d and Supplementary Fig. 7b**). Moreover, absolute quantitative real-time PCR assay of pri-miR146a showed NDR1-deficiency enhanced the copy number of pri-miR146a (**Fig. 7a**), while overexpression of NDR1 inhibited pri-miR146a transcription (**Fig. 7b**). Together, these results suggest that NDR1 decreases miR146a expression by dampening its transcription, which is conserved between mice and humans.

Reference

1. Kieffer-Kwon KR, *et al.* Interactome maps of mouse gene regulatory domains reveal basic principles of transcriptional regulation. *Cell* **155**, 1507-1520 (2013).
2. Nobrega MA, Ovcharenko I, Afzal V, Rubin EM. Scanning human gene deserts for long-range enhancers. *Science* **302**, 413 (2003).
3. Hou J, *et al.* MicroRNA-146a feedback inhibits RIG-I-dependent Type I IFN production in macrophages by targeting TRAF6, IRAK1, and IRAK2. *Journal of immunology* **183**, 2150-2158 (2009).
4. Ghose J, Bhattacharyya NP. Transcriptional regulation of microRNA-100, -146a, and -150 genes by p53 and NFkappaB p65/RelA in mouse striatal STHdh(Q7)/ Hdh(Q7) cells and human cervical carcinoma HeLa cells. *RNA biology* **12**, 457-477 (2015).
5. Taganov KD, Boldin MP, Chang KJ, Baltimore D. NF-kappaB-dependent induction of microRNA miR-146, an inhibitor targeted to signaling proteins of innate immune responses. *Proceedings of the National Academy of Sciences of the United States of America* **103**, 12481-12486 (2006).
6. Schneider WM, Chevillotte MD, Rice CM. Interferon-Stimulated Genes: A Complex Web of Host Defenses. *Annual Review of Immunology, Vol 32* **32**, 513-545 (2014).
7. Santos-Rosa H, *et al.* Active genes are tri-methylated at K4 of histone H3. *Nature* **419**, 407-411 (2002).
8. Cao R, Wang H, He J, Erdjument-Bromage H, Tempst P, Zhang Y. Role of hPHF1 in H3K27 methylation and Hox gene silencing. *Molecular and cellular biology* **28**, 1862-1872 (2008).
9. Cornils H, Kohler RS, Hergovich A, Hemmings BA. Human NDR Kinases Control G(1)/S Cell Cycle Transition by Directly Regulating p21 Stability. *Molecular and cellular biology* **31**, 1382-1395 (2011).
10. Joffre C, *et al.* The Pro-apoptotic STK38 Kinase Is a New Beclin1 Partner Positively Regulating Autophagy. *Curr Biol* **25**, 2479-2492 (2015).
11. Ma CM, *et al.* NDR1 protein kinase promotes IL-17-and TNF-alpha-mediated inflammation by competitively binding TRAF3. *EMBO reports* **18**, 586-602 (2017).

Reviewers' comments:

Reviewer #1 (Remarks to the Author):

The revision does improve the manuscript. I would have the following comments to add:

1. In the revised manuscript the authors provided new data on the interference of NF- κ B-mediated regulation of miR-146a expression by NDR1 in human THP-1 cells, which looked convincing. They also added a new citation that had reported the location of a NF- κ B regulatory region ~10kb upstream of mouse miR-146a. This provided rationality for their examination of the region. It is still not clear if binding of NDR1 at the intergenic region, which is ~60kb away from NF- κ B binding site, is required to decrease the latter binding, or other mechanism may exist, and the intergenic association is an independent event. Such possibility should be investigated .

2. Viral infection inhibits NDR1 expression in STAT1-dependent manner. H3K27me3 is up-regulated in NDR1 promoter after viral infection. Authors may provide whether this phenomena occurs after IFN stimulation and STAT1 binds to NDR1 promoter.

3. Line 215 in Page 11: The conclusion "NDR1 has no effect on VSV induced-IRF3 activation" is not convincing only based on reporter gene assay.

4. Regarding quantification of mature miR-146a, the authors didn't state in the previous manuscript they were using stem-loop RT-qPCR, which was confusing. According to "STEM-LOOP RT-qPCR for miRNA" (Kramer M, Curr Protoc Mol Biol. 2011 Jul; CHAPTER: Unit15.10.), a proper hydrolysis probe should be included to achieve specificity. It's noteworthy that the authors now also did Northern blot for miR-146a to address the miRNA processing, which confirmed the changes in mature miR-146a levels caused by NDR1. Yet, for the qPCR step, at least the primer sequence should be provided if no probe was used.

5. Line 372 in page19: "probe for mouse miR-146a are not available" served as the excuse for not performing northern blot in mouse, which is not correct. The mature sequence of miR-146a of mouse is similar to human.

```
>hsa-miR-146a-5p MIMAT0000449
```

```
UGAGAACUGAAUCCAUGGGUU
```

```
>mmu-miR-146a-5p MIMAT0000158
```

```
UGAGAACUGAAUCCAUGGGUU
```

Probe for testing miR-146a mature sequence can show both mature miR-146a and precursor miR-146a through northern blot assay. Author could improve northern blot system to detect precursor miR-146a.

6. NDR1 significantly inhibited miR-146a expression, but had no effect on expression of IRAK1 and TRAF6, which are the important target of miR-146a. The explanation of this result is not convincing.

7. NDR1 inhibits miR146a expression through NF κ B-mediated transcription. STAT1 involved in this process. However, author claimed that NDR1 regulated miR146a expression in STAT1-independent manner. The logic is confused and does not make sense.

Reviewer #2 (Remarks to the Author):

Overall, my concerns have been adequately addressed. I still do not see where a description of how the miR-146a -/- 293T and L929 cells were developed or obtained in the methods and this should be added.

To Reviewer 1:

Thank you very much for your comments and suggestions regarding our manuscript entitled “Downregulated NDR1 protein kinase inhibits innate immune response by initiating an miR146a-STAT1 feedback loop” (NCOMMS-17-12910A). According to your suggestions, we have added some new data and re-organized the manuscript. If you have any further questions, please inform us and we can discuss them further.

The revision does improve the manuscript. I would have the following comments to add:

1. Question: *In the revised manuscript the authors provided new data on the interference of NF- κ B-mediated regulation of miR-146a expression by NDR1 in human THP-1 cells, which looked convincing. They also added a new citation that had reported the location of a NF- κ B regulatory region ~10kb upstream of mouse miR-146a. This provided rationality for their examination of the region. It is still not clear if binding of NDR1 at the intergenic region, which is ~60kb away from NF- κ B binding site, is required to decrease the latter binding, or other mechanism may exist, and the intergenic association is an independent event. Such possibility should be investigated.*

Response: Yes, you are right. In order to explore whether the NDR1 inhibits NF- κ B mediated miR146a expression via binding at the miR146a intergenic region. We had designed 18 gRNAs (7 gRNAs at the first revision (**Response Figure 1**) and another 11 gRNAs at the second revision (**Response Figure 3**)) to obtain miR146a intergenic region deficient cells using the MIT online tool (<http://crispr.mit.edu/>). Unfortunately, none of the constructed CRISPR-cas9 plasmids generated double strand breaks (DSBs) in miR146a intergenic region (**Response Figure 2,4**). Using the same CRISPR-cas9 system, we had obtained NDR1 and miR146a deficient L929 cell lines, indicating that this CRISPR-cas9 system does work well to generate a knockout. As reported by Peng R et al, once the site-specific double-strand breaks (DSBs) of DNA are created by Cas9 nuclease, two different repair mechanisms can be activated, including the non-homologous end joining (NHEJ) and the homology-directed repair (HDR)

mechanisms. In the error-free HDR pathway, the DNA template used for repair is identical to the original DNA sequence at the DSBs¹, which could result in the failure of gene knockout. It could be the HDR pathway is activated by DSBs of DNA in miR146a intergenic region. Alternatively, some special DNA modifications in miR146a intergenic region may inhibit the occurrence of DSBs caused by Cas9 nuclease².

```

AAAAAGAGGGCTCCTCATGCCTGGGATTGGGTTTCTGTTAGACAGTATTTGGCTCTTGGAAAGGAGCATTGTCAACCCAG
TGAGGAACATTTTTAAATGTTGGAAATAAAAAATATGTCAAATAAAAAACACCAAACCATAAGCAATAGATTACGCCAGAC
CGAAGGAAAAGTACCAGAGATAGAGGACATGGATGCGGAAATACTACACACAAAACACCGATAAATAAACTAAATAAAC
ATGACTGCAATGTCCAAGAATCCTGAGATACAATCAAAAAACAAAAGCAGAGTCCAAAAGACAAAAGGAACTAAGAAT
TTTTTAAAGGACAGACAGAGAGAATGTATTATTCAATAAATTAAGAACAAAATAGACAGGTTAGAGGGTGCGCCAGAGA
ACCGGACAGCTTCTGGGACGGGCGGAAGCACAGAGCCGCTGAGGCAGCACCTTGGCGGGCCGCAGACAACCGGCCACC
ATCCGGACCAGAGGACAGGTTGTCTGCCTGGCTTGGGAGGCGGCCCTCAGCCTCAGCAGCAGCGGTCGCCATCTTGTTCCG
GGACTCAGCAGAACTGGGAAATTAGTCTGAACAGGTTAGAGGGTGCGCCAGAGAACCGGACAGCTTCTGGGACGGGCAG
AAGCACAGAGCCGCTGAGGCAGCACCTTGGCGGGCCGCAGACAGCCGGCCACCGTCTGGACCAGAGGACAGGTGCCCG
CCTGGCTTGGGAGGCGGCCTCAGCCTCAGCAGCAGCGGTTGCCATCTGGGTTCCAGGA

```

Response Figure 1. The sequence of murine miR146a intergenic region. The NDR1 DNA binding site is highlighted; the gRNAs targeted sequences were underlined, the primers for genotype identification are marked in green.

Response Figure 2. PCR analysis of genomic DNA from L929 cells transfected with CRISPR-cas9 plasmids containing gRNAs to target miR146a intergenic region.

AAGCCCAAAGACTTGCAAAGCCATCTGTGTGCACTGGTCATTGTTGCTGTTGTTCTTAGGTGTCATAGCTGGGCAAGGCTG
 TTGGTTGTATCCTTCTTCTGATGCCATAAAACACTGGTCTCAAGGAGGAGGCTTCCGGTCAATCCATCCCAGCAATCTA
 AATGCATGGTGCACCTTCAAGGGGGACAGCCAGGGCCAACAGAAATAGCCTGTAGTGTTTTGGGCAITTCCTGGATGACC
 ATTATCAACGACTCAAAAAGAGGGGCTCCTCATGCTGGGATTGGGGTTCTGTTAGACAGTATTTGGCTCTTGGAAAGGAGC
6#
ATTGTCAACCCAGTGAGGAAACATTTTTAAATGTTGGAAATAAAAAATATGTCAAATAAAAAACACCAAACCATAAGCAAT
 AGATTAGCCAGACCGAAGGAAAAGTACCAGAGATAGAGGACATGGATGCGGAAATACTACACACAAAACCCGATAAAT
 AAAACTAAATAAACATGACTGCAATGTCCAAGAATCCTGAGATACAATCAAAAAACAAAAGCAGAGTCCAAAAGACAA
 AAGGAATAAGAATTTTTTAAAGGACAGACAGAGAGAATGTATTATTCAATAAATAAGAACAAAATAGACAGGTTAGA
 GGGTGCACCAGAGAACCGGACAGCTTCTGGGACGGCGGAAGCACAGAGCCGCTGAGGCAGCACCCCTTGCGGGCCGCA
GACAACCGGCCACCATCCGGACCAGAGGACAGGTGTCTGCCTGGCTTGGGAGGCGGCCCTCAGCCTCAGCAGCAGCGGTCC
 CCATCTGGTCCGGGACTCAGCAGAACTGGGAAATTAGTCTGAACAGGTTAGAGGGTGCGCCAGAGAACCGGACAGCTT
7#
CTGGGACGGGCAGAAAGCACAGAGCCGCTGAGGCAGCACCCCTTGGCGGGCCGCAGACAGCCGGCCACCGCTGGACCAGA
8#
GGACAGGTGCCCGCTGGCTTGGGAGGCGGCCTCAGCCTCAGCAGCAGCGGTTGCCATCTGGGTTCAGGACTCCGCGGG
10#
11#
 ACCTAGGAAATTAGTCTGAACAGGTTAGAGGGTGCGCCAGAGAACCGGACAGCTTCTGGGACAGCCGGAAGCACAGAGC
 CGCTGAGGCAGTACCCTTTGCAGGCTGCAGACAGCCGGCCACCGTCCAGACCAGAGGACAGGTATCCGCTGGCTCGGGA

Response Figure 3. The sequence of murine miR146a intergenic region. The NDR1 DNA binding site is highlighted; the gRNAs targeted sequences were underlined, the primers for genotype identification are marked in green.

Response Figure 4. PCR analysis of genomic DNA from L929 cells transfected with CRISPR-cas9 plasmids containing gRNAs to target miR146a intergenic region.

2. Question: *Viral infection inhibits NDR1 expression in STAT1-dependent manner. H3K27me3 is up-regulated in NDR1 promoter after viral infection. Authors may provide whether this phenomena occurs after IFN stimulation and STAT1 binds to NDR1 promoter.*

Response: Thanks for your suggestion. Thp1 cells were treated with recombinant human IFN α . As shown in Fig. 1e and Supplementary Fig. 1e, IFN α stimulation significantly decreased NDR1 expression at both the mRNA and protein levels. By ChIP assay, we found that H3K27me3 was upregulated in the promoter of NDR1 gene in Thp1 cells after VSV infection or IFN α stimulation, whereas H3K4me3 remained constant throughout the infection with VSV or IFN α stimulation (Fig. 1h,i). Together, these data indicate that viral infection downregulates NDR1 expression via an IFN signaling-dependent mechanism and may also involve epigenetic modification of histone H3.

We found a putative STAT1 binding site (TTACTTTAA) in the human NDR1 promoter region (Response Figure 5) and ChIP assay was performed to test whether STAT1 was able to bind to NDR1 promoter. As shown in Supplementary Fig. 1h, STAT1 bound to NDR1 promoter region in an IFN α -stimulation-dependent way, indicating that STAT1 may act as a negative transcriptional regulator for NDR1 transcription during IFN α stimulation.

```
GGGCTTTGAAGATGAAATGAATGAATAACTCGCAGCACATGCTTCCTTCTCTCCTAGATGGCCCTTTGCAGGCAGTCAG
GGACCATGTCTCGTTCATCTTAGTTTCCAGGAGCTTGACACTTGTAGTTGAGGGCTTTGTGCTCCATTCTCAAGAAACGAC
AAGTGCCCAAAGGCTATTAATACTTGGGACCTCTTTACCCCAAAAGTAGTACAGAATAAGAGTTGGGTGGCCTTGAAAAG
CACCTTCTTGTGCCACACTGGGTCGCGCTGGGTAACCTCCCAAACCTATGAGAGCTCACCTTATCCCAGGTAAGGTCCCT
TACTTTAAGGAAGAGGAGGGGGCCAAAGGGCTCCAGCAGACCTGTCTACACGGGGTAGGGGGTGGGTGGGAACTATA
CAGGTGACAACATTTTCATAGAACTACACACACTCACACACACGCACATGACTGCATACCCAACAGGTTAAATCCGAGT
GTGGTCTGTAATGAAGTTAATTGTATTGGTTAGTTAATTGTATTGGTTAGTTAATTGTATTGTGCGAATGTCAATTTCTTG
GCTTTGAGAATGTTGTATCTTAAGAAAAATGTTGAGGTAAGCTGGGTGGGTGATGGTATAGGGAACCTGTATGTATTAATT
TTGCAACTTCTGTGAGTCCATAATTATTTGCAAGTAAAAAGTAAAACAAAAGAAAACAATGGAACCTCCCCCTCAGCCCA
GCCTACTTCGAGACCGCCTCACACTCACACATCTGGCCTCACCCAGGTCCAGCCAGCCCTAGGCAGGGGGTGAAGGGAGG
GGCAGGGCGGGGCCACGCAAGCGCAGTTCGCAATCCCGGCCGCTCCGGGAAGTCGCG
```

Response Figure 5. The sequence of human NDR1 promoter region. The putative STAT1 DNA binding site is highlighted; the primers for ChIP-qPCR are marked in green.

3. Question: Line 215 in Page 11: The conclusion “NDR1 has no effect on VSV induced-IRF3 activation” is not convincing only based on reporter gene assay.

Response: Sorry for the confusion. In our study, reporter assay showed that NDR1

had no effect on RIG-I-N-induced IRF3 reporter activation (Supplementary Fig. 4c). NDR1 knockout (Fig. 4a) or knockdown (Supplementary Fig. 4a) had no effect on the activation of IRF3 pathways and the expression of RIG-I in peritoneal macrophages upon VSV infection for 4 h and slightly inhibited the VSV-induced the activation of the IRF3 pathways and the expression of RIG-I at the 8 hours after VSV infection. Moreover, NDR1 deficiency significantly attenuated STAT1 expression (Supplementary Fig. 5c) and the promoting effect of NDR1 on anti-virus immune response was abolished in STAT1-deficient macrophages (Fig. 5c,d). Taken together, these data indicate that NDR1 has no direct effect on VSV induced-IRF3 activation, but enhances STAT1 expression and promotes ISGs induction including RIG-I, which further facilitates the activation of the IRF3 (Fig. 4b) at the 8 hours after VSV infection. The conclusion “NDR1 has no effect on VSV induced-IRF3 activation” is indeed confusing. We have changed this sentence into “NDR1 has no direct effect on VSV induced-IRF3 activation” in the current manuscript.

4. Question: *Regarding quantification of mature miR-146a, the authors didn't state in the previous manuscript they were using stem-loop RT-qPCR, which was confusing. According to “STEM-LOOP RT-qPCR for miRNA” (Kramer M, Curr Protoc Mol Biol. 2011 Jul; CHAPTER: Unit15.10.), a proper hydrolysis probe should be included to achieve specificity. It's noteworthy that the authors now also did Northern blot for miR-146a to address the miRNA processing, which confirmed the changes in mature miR-146a levels caused by NDR1. Yet, for the qPCR step, at least the primer sequence should be provided if no probe was used.*

Response: Sorry for the mistake and thanks for your suggestion. The primer sequence for mature miR146a was listed in Table S4. In the revised manuscript, TaqMan microRNA assay using specific hydrolysis probes for mature miR146a and U6 was performed to achieve specificity. As shown in Fig. 7c and Supplementary Fig. 7a, NDR1 knockout in mouse PMs or knockdown in human PBMCs enhanced mature miR146a expression. The overexpression of NDR1 in RAW264.7 (Fig. 7d) or Thp1 cells (Supplementary Fig. 7b) inhibited mature miR146a expression. In addition,

silencing NDR1 enhanced mature miR146a expression in various kinds of cells (Fig. 7e).

5. Question: Line 372 in page19: “probe for mouse miR-146a are not available” served as the excuse for not performing northern blot in mouse, which is not correct. The mature sequence of miR-146a of mouse is similar to human.

>hsa-miR-146a-5p MIMAT0000449 :UGAGAACUGAAUCCAUGGGUU

>mmu-miR-146a-5p MIMAT0000158 : UGAGAACUGAAUCCAUGGGUU

Probe for testing miR-146a mature sequence can show both mature miR-146a and precursor miR-146a through northern blot assay. Author could improve northern blot system to detect precursor miR-146a.

Response: Sorry for the mistake. We have replaced the sentence with “Considering that the radiative facility is not available, we used the digoxigenin system to examine the miR146a processing in Thp1 cells.” in the current manuscript.

We tried to improve the northern blot system by optimizing the temperature of hybridization, increasing the concentration of anti-DIG antibody, prolonging the time of antibody incubation and increasing the loading amount of the samples. As shown in Response Figure. 6, only the expression of mature miR146a but not the expression of pri- and pre-miR146a was detectable in Thp1 cell even at the dose of 500 µg. Meanwhile, none of pri- miR146a, pre-miR146a and mature miR146a expression but the expression of the internal control U6 was detectable in mouse PMs. RNA showed obscure strip in blot at the dose from 200 µg to 500 µg because the amount of RNA samples was “overloaded”. Taken together, these data indicate that the expression of pre- and pri-miR146a in Thp1 cells and the abundance of pri-, pre-miR146a and mature miR146a in mouse macrophages were too low to be detected by the digoxigenin system.

Response Figure 6. Northern blot analysis of miR146a expression in PMs and Thp1 cells. The increasing amounts of the PMs or Thp1 RNA samples were loaded on 17% denaturing PAGE (dPAGE) for Northern blot analysis.

6. Question: *NDR1 significantly inhibited miR-146a expression, but had no effect on expression of IRAK1 and TRAF6, which are the important target of miR-146a. The explanation of this result is not convincing.*

Response: NDR1 deficiency attenuated the expression of STAT1 but had no effect on IRAK1 and TRAF6 expression (**Fig. 7l**). We explored the sensitivity of STAT1, IRAK1 and TRAF6 to miR146a-mediated inhibition in the current manuscript. HEK293 cells were transfected with pre-miR146a overexpressing plasmid, which mimics the natural expression of miR146a in the cells. As shown in **Fig. 7m**, transfecting low dose (2 ng) of pre-miR146a plasmid significantly attenuated STAT1 expression, while at least transfecting 20 ng of pre-miR146a plasmid could inhibit the expression of IRAK1 and TRAF6, indicating that STAT1 is more sensitive to miR146a-mediated inhibition than IRAK1 and TRAF6. Real-time PCR showed that 2

ng of pre-miR146a plasmid transfection resulted in about 2-fold more miR146a expression in HEK293 cells compared to the control transfectants (Supplementary Fig. 7t). In addition, NDR1 deficient macrophage expressed about 2-fold more miR146a than WT macrophage in the resting state (Fig. 7c). We found that STAT1 also binds to the miR146a promoter and inhibits miR146a expression. Higher susceptibility of STAT1 to miR146a-mediated inhibition and the mutual inhibition between miR146a and STAT1 may be the underlying mechanism by which NDR1 specifically upregulates STAT1 but does not affect the expression of IRAK1 or TRAF6 in the resting state.

7. Question: NDR1 inhibits miR146a expression through NF-κB-mediated transcription. STAT1 involved in this process. However, author claimed that NDR1 regulated miR146a expression in STAT1-independent manner. The logic is confused and does not make sense.

Response: In this study, we demonstrated that NDR1 directly binds to the intergenic region of miR146a and inhibits NF-κB-mediated miR146a expression (Fig. 6g and Fig. 7a,b). NDR1 silencing enhanced miR146a expression in both WT and *STAT1*^{-/-} macrophages (Fig. 8g), indicating that NDR1 regulates miR146a expression in a STAT1-independent manner at the basal level. However, upon virus infection, reduced NDR1 facilitates NF-κB-mediated miR146a transcription which leads to the inhibition of STAT1 translation. Then decreased STAT1 promotes miR146a expression via the feedback loop between STAT1 and miR146a. Therefore, NDR1 regulates miR146a expression in a STAT1-dependent manner during virus infection. The conclusion “NDR1 inhibits miR146a expression in a STAT1-independent manner” is indeed confusing. We have changed this sentence into “NDR1 inhibits miR146a expression in a STAT1-independent manner at the basal level” in the current manuscript.

To Reviewer 2:

Thank you very much for your comments and suggestions regarding our manuscript entitled “Downregulated NDR1 protein kinase inhibits innate immune response by initiating an miR146a-STAT1 feedback loop” (NCOMMS-17-12910). According to your suggestions, we have re-organized the manuscript. If you have any further questions, please inform us and we can discuss them further.

1. Question: Overall, my concerns have been adequately addressed. I still do not see where a description of how the miR-146a -/- 293T and L929 cells were developed or obtained in the methods and this should be added.

Response: Thanks for your suggestion. We have added the description of how the miR-146a -/- 293T and L929 cells were developed in the methods section on page 29 in the current supplementary as bellows: “To generate miR146a-KO cells, we designed two gRNAs to target the upstream and downstream flanking genomic sequence of pre-miR146a respectively, and cloned them in the CRISPR-cas9 plasmids pEP-330x, which were then co-transfected into HEK293 or L929 cells using PEI (Polysciences, cat# 23966-2) for 48 h”

Response Reference

1. Peng R, Lin G, Li J. Potential pitfalls of CRISPR/Cas9-mediated genome editing. *The FEBS journal* **283**, 1218-1231 (2016).
2. Bryson AL, *et al.* Covalent Modification of Bacteriophage T4 DNA Inhibits CRISPR-Cas9. *mBio* **6**, e00648 (2015).

Reviewers' comments:

Reviewer #1 (Remarks to the Author):

Authors have tried very hard to address the concerns by adding new experiment data and some further explanations but I still can not be convinced to accept these major conclusions based on current available data . Critical defects for this study are the molecular mechanisms proposed for explaining the phenotypes linked to NDR1 deficiency . I can recognize the biological importance of NDR1 deficiency but I strongly suggest authors could do further comprehensive experiments to generate reliable mechanisms and insights for the biological functions of NDR1 on the regulation of Innate immunity .

My major concerns are followed :

Authors persisted to argue NDR1 can mediate the pathway by only inhibiting miR146a expression . but on second revision there is no solid data to support NDR1 can do that by acting at the intergenic region of miR146a and showed unusual effects on regulating the well established miR146 targets . Authors failed on several CRISPR based genome editing experiments . It could be alternatively to use CRISPR interference (dCas9 system) to explore this possible role . Recently there are many elegant papers demonstrated this novel approach is very powerful to define the distal gene expression regulation.

Most important issue that authors should clarify is to exclude other protein(s) or non-coding RNAs molecule(s) could be more important for NDR1 as its functional cooperators. The integrative genomics ,proteomics and epigenomics analysis approach can provide unbiased clues for molecular mechanisms linked to NDR1 deficiency. Authors should apply these new approaches to address critical scientific questions.

To the Reviewer:

Thank you very much for your comments and suggestions regarding our manuscript entitled “Downregulated NDR1 protein kinase inhibits innate immune response by initiating an miR146a-STAT1 feedback loop” (NCOMMS-17-12910B). According to your suggestions, we have added some new data and re-organized the manuscript. If you have any further questions, please inform us and we can discuss them further.

Authors have tried very hard to address the concerns by adding new experiment data and some further explanations but I still can not be convinced to accept these major conclusions based on current available data. Critical defects for this study are the molecular mechanisms proposed for explaining the phenotypes linked to NDR1 deficiency. I can recognize the biological importance of NDR1 deficiency but I strongly suggest authors could do further comprehensive experiments to generate reliable mechanisms and insights for the biological functions of NDR1 on the regulation of Innate immunity.

My major concerns are followed :

1. Question: *Authors persisted to argue NDR1 can mediate the pathway by only inhibiting miR146a expression. But on second revision there is no solid data to support NDR1 can do that by acting at the intergenic region of miR146a and showed unusual effects on regulating the well established miR146 targets. Authors failed on several CRISPR based genome editing experiments. It could be alternatively to use CRISPR interference (dCas9 system) to explore this possible role. Recently there are many elegant papers demonstrated this novel approach is very powerful to define the distal gene expression regulation.*

Most important issue that authors should clarify is to exclude other protein(s) or non-coding RNAs molecule(s) could be more important for NDR1 as its functional cooperators. The integrative genomics, proteomics and epigenomics analysis approach can provide unbiased clues for molecular mechanisms linked to NDR1 deficiency. Authors should apply these new approaches to address critical scientific questions.

Response: Thanks for your suggestions. In order to explore whether NDR1 decreases NF- κ B-mediated miR146a transcription depending on the binding of NDR1 to the intergenic region of miR146a, we designed 12 gRNAs targeting the negative strand of miR146a intergenic region to generate NDR1 DNA binding site deficient (NDR1 DBS^{-/-}) L929 cells (**Response Figure 1**). As shown in **Response Figure 2**, 6# and 11# CRISPR-cas9 plasmids co-transfection generated double strand breaks (DSBs) in miR146a intergenic region. The NDR1 DBS^{-/-} L929 cells were validated at DNA level by PCR (**Supplementary Fig. 7n**).

To investigate whether NDR1 decreases NF- κ B-mediated miR146a transcription depending on its binding to the intergenic region of miR146a, we transfected flag-NDR1 or empty vector into *NDR1* DBS^{+/+} and *NDR1* DBS^{-/-} L929 cells. As shown in **Fig. 7d**, overexpressing NDR1 attenuated the enrichment of NF- κ B at the miR146a regulatory sequences in *NDR1* DBS^{+/+} cells but this effect was abolished in *NDR1* DBS^{-/-} L929 cells. Moreover, the effects of NDR1 on the expression of miR146a, STAT1 and viral-induced ISGs were also repealed in *NDR1* DBS^{-/-} L929 cells (**Fig. 7e,f** and **Supplementary Fig. 7n**). Taken together, these results indicate that NDR1 dampens NF- κ B-mediated miR146a transcription in an miR146a intergenic region-binding dependent manner.

Even though the antiviral effect of NDR1 is abolished in miR146a- or NDR1 DBS-deficient cells, we cannot exclude the possibility that other protein(s) or non-coding RNAs molecule(s) could be important for NDR1 as its functional cooperators. The integrative genomics, proteomics and epigenomics analysis approach could be further used to explore this possibility. We have added the above description in the discussion section on page 26 in the current manuscript.

GACGGTGGCCGGCTGTCTGCAGCCTGCAAAGGGTACTGCCTCAGCGGCTCTGTGCTTCCGCTGTCCCAGA
 AGCTGTCCGGTTCTCTGGCGCACCTCTAACCTGTTCAGACTAATTCCTAGGTCCCGCGGAGTCCTGGAAC
CCAGATGGCAACCGCTGCTGCTGAGGCTGAGGCCGCTCCCAAGCCAGGCGGGCACCTGTCTCTGGTCC
 AGACGGTGGCCGGCTGTCTGCGGCCCGCAAGGGTGTGCCTCAGCGGCTCTGTGCTTCTGCCGTCCCAG
 AAGCTGTCCGGTTCTCTGGCGCACCTCTAACCTGTTCAGACTAATTCCAGTTCTGCTGAGTCCCGGAACC
AAGATGGCGACCGCTGCTGCTGAGGCTGAGGCCGCTCCCAAGCCAGGCAGACACCTGTCTCTGGTCCG
 GATGGTGGCCGGTTGCTGCGGGCCCGCAAGGGTGTGCCTCAGCGGCTCTGTGCTTCCGCCGTCCCAGA
 AGCTGTCCGGTTCTCTGGCGCACCTCTAACCTGTCTATTTGTTCTTAATTTATTGAATAATACATTCTCTCTG
 TCTGTCTTTAAAAAATTCTTAGTTCCTTTTGTCTTTGGACTCTGCTTTTGTTTTTTTGATTGATCTCAGGA
 TTCTTGACATTGCAGTCATGTTTATTAGTTTATTATCGGTGTTTGTGTAGTATTTCCGCATCCATGTC
TCTATCTCTGGTACTTTTCCTTCGGTCTGGCTGAATCTATTGCTTATGGTTTGGTGTTTTTATTGACATATTTT
 TATTCCAACATTTAAAAATGTTCTCACTGGGGTTGACAATGCTCCTTCCAAGAGCCAAATACTGTCTAACA
 GAAACCCCAATCCCAGGCATGAGGAGCCCTCTTTTGAGTCGTTGATAATGGTCATCCAGGAAATGCCCAA
ACACTACAGGCTATTTCTGTTGGCCCTGCTGCCCTTGAAGGTGCACCATGCATTTAGATTGCTGGGATG
 GAATTGACCGAAAGCCTCCTCCTTGAGAACCAGTGTATGGCATCAGAAGGAAGGATACAACCAACAGC
 CTTGCCAGCTATGACACCTAAGAACAACAGCAACAATGACCAGTGACACAGATGGCTTTGCAAGTCTTTG
 GGCTTGGCTGGGTATCTCGCCCTCCTCTGACTCCCCAGGTTATATTCCAAGGGTGTTAATGAATGTCCAG

Response Figure 1. The sequence of the negative chain for the murine miR146a intergenic region. The NDR1 DNA binding site is highlighted; the gRNAs targeted sequences were underlined or marked in blue, the primers for genotype identification are marked in green.

Response Figure 2. PCR analysis of genomic DNA from L929 cells transfected with CRISPR-cas9 plasmids containing gRNAs to target miR146a intergenic region.

Reviewers' comments:

Reviewer #1 (Remarks to the Author):

For this second revision , Authors did not address my major concerns properly.
They did not do the experiments I have suggested to overcome critical limitations
I also have not seen any resonable arguments for my critiques.
I still would suggest authors could do further experiments to support their major conclusions.

To the Reviewer:

Thank you very much for your comments and suggestions regarding our manuscript entitled “Downregulated NDR1 protein kinase inhibits innate immune response by initiating an miR146a-STAT1 feedback loop” (NCOMMS-17-12910B). According to your suggestions, we have added some new data and re-organized the manuscript. If you have any further questions, please inform us and we can discuss them further.

Authors have tried very hard to address the concerns by adding new experiment data and some further explanations but I still can not be convinced to accept these major conclusions based on current available data. Critical defects for this study are the molecular mechanisms proposed for explaining the phenotypes linked to NDR1 deficiency. I can recognize the biological importance of NDR1 deficiency but I strongly suggest authors could do further comprehensive experiments to generate reliable mechanisms and insights for the biological functions of NDR1 on the regulation of Innate immunity.

My major concerns are followed :

1. Question: *Authors persisted to argue NDR1 can mediate the pathway by only inhibiting miR146a expression. But on second revision there is no solid data to support NDR1 can do that by acting at the intergenic region of miR146a and showed unusual effects on regulating the well established miR146 targets. Authors failed on several CRISPR based genome editing experiments. It could be alternatively to use CRISPR interference (dCas9 system) to explore this possible role. Recently there are many elegant papers demonstrated this novel approach is very powerful to define the distal gene expression regulation.*

Most important issue that authors should clarify is to exclude other protein(s) or non-coding RNAs molecule(s) could be more important for NDR1 as its functional cooperators. The integrative genomics, proteomics and epigenomics analysis approach can provide unbiased clues for molecular mechanisms linked to NDR1 deficiency. Authors should apply these new approaches to address critical scientific questions.

Response: Thanks for your suggestions. In order to explore whether NDR1 decreases NF- κ B-mediated miR146a transcription depending on the binding of NDR1 to the intergenic region of miR146a, we designed 12 gRNAs targeting the negative strand of miR146a intergenic region to generate NDR1 DNA binding site deficient (NDR1 DBS^{-/-}) L929 cells (**Response Figure 1**). As shown in **Response Figure 2**, 6# and 11# CRISPR-cas9 plasmids co-transfection generated double strand breaks (DSBs) in miR146a intergenic region. The NDR1 DBS^{-/-} L929 cells were validated at DNA level by PCR (**Supplementary Fig. 7n**).

To investigate whether NDR1 decreases NF- κ B-mediated miR146a transcription depending on its binding to the intergenic region of miR146a, we transfected flag-NDR1 or empty vector into *NDR1* DBS^{+/+} and *NDR1* DBS^{-/-} L929 cells. As shown in **Fig. 7d**, overexpressing NDR1 attenuated the enrichment of NF- κ B at the miR146a regulatory sequences in *NDR1* DBS^{+/+} cells but this effect was abolished in *NDR1* DBS^{-/-} L929 cells. Moreover, the effects of NDR1 on the expression of miR146a, STAT1 and viral-induced ISGs were also repealed in *NDR1* DBS^{-/-} L929 cells (**Fig. 7e,f** and **Supplementary Fig. 7n**). Taken together, these results indicate that NDR1 dampens NF- κ B-mediated miR146a transcription in an miR146a intergenic region-binding dependent manner.

Even though the antiviral effect of NDR1 is abolished in miR146a- or NDR1 DBS-deficient cells, we cannot exclude the possibility that other protein(s) or non-coding RNAs molecule(s) could be important for NDR1 as its functional cooperators. The integrative genomics, proteomics and epigenomics analysis approach could be further used to explore this possibility. We have added the above description in the discussion section on page 26 in the current manuscript.

GACGGTGGCCGGCTGTCTGCAGCCTGCAAAGGGTACTGCCTCAGCGGCTCTGTGCTTCCGCTGTCCCAGA
AGCTGTCCGGTTCTCTGGCGCACCTCTAACCTGTTCAGACTAATTTCTAGTCCCGCGGAGTCCTGGAAC
CCAGATGGCAACCGCTGCTGCTGAGGCTGAGGCCGCTCCCAAGCCAGGCGGGCACCTGTCTCTGGTCC
AGACGGTGGCCGGCTGTCTGCGGCCCGCAAGGGTGTGCCTCAGCGGCTCTGTGCTTCTGCCGTCCCAG
AAGCTGTCCGGTTCTCTGGCGCACCTCTAACCTGTTCAGACTAATTTCCAGTTCTGCTGAGTCCCGGAACC
AAGATGGCGACCGCTGCTGCTGAGGCTGAGGCCGCTCCCAAGCCAGGCGAGACACCTGTCTCTGGTCCG
GATGGTGGCCGGTGTCTGCAGCCTGCAAAGGGTGTGCCTCAGCGGCTCTGTGCTTCCGCCCCTCCCAGA
AGCTGTCCGGTTCTCTGGCGCACCTCTAACCTGTCTATTTTGTCTTAATTTATTGAATAATACATTCTCTCTG
TCTGTCTTTAAAAAATTCTTAGTTCCTTTTGTCTTTGGACTCTGCTTTTGTTTTTTTGATTGATCTCAGGA
TTCTTGACATTGCAGTCATGTTTATTAGTTTTATTATCGGTGTTTGTGTAGTATTTCCGATCCATGTCC
TCTATCTCTGGTACTTTTCCTTCGGTCTGGCTGAATCTATTGCTTATGGTTTGGTGTTTTTATTGACATATTTT
TATTCCAACATTTAAAAATGTTCTCACTGGGGTTGACAATGCTCCTTCCAAGAGCCAAATACTGTCTAACA
GAAACCCCAATCCCAGGCATGAGGAGCCCTCTTTTGAGTCGTTGATAATGGTCATCCAGGAAATGCCCAA
ACACTACAGGCTATTTCTGTTGGCCCTGCTGCCCTTGAAGGTGCACCATGCATTTAGATTGCTGGGATG
GAATTGACCGAAAGCCTCCTCCTTGAGAACCAGTGTATGGCATCAGAAGGAAGGATAACAACCAACAGC
CTTGCCAGCTATGACACCTAAGAACAACAGCAACAATGACCAGTGCACACAGATGGCTTTGCAAGTCTTTG
GGCTTGGCTGGGTATCTCGCCCTCCTCTGACTCCCCAGGTTATATTCCAAGGGTGTTAATGAATGTCCAG

Response Figure 1. The sequence of the negative chain for the murine miR146a intergenic region. The NDR1 DNA binding site is highlighted; the gRNAs targeted sequences were underlined or marked in blue, the primers for genotype identification are marked in green.

Response Figure 2. PCR analysis of genomic DNA from L929 cells transfected with CRISPR-cas9 plasmids containing gRNAs to target miR146a intergenic region.